# DOP: Off-Policy Multi-Agent Decomposed Policy Gradients

**Yihan Wang**[*], **Beining Han**[*], **Tonghan Wang**[*], **Heng Dong, Chongjie Zhang**
Institute for Interdisciplinary Information Sciences
Tsinghua University, Beijing, China
{memoryslices,bouldinghan,tonghanwang1996,drdhii}@gmail.com
chongjie@tsinghua.edu.cn

## Abstract

Multi-agent policy gradient (MAPG) methods recently witness vigorous progress. However, there is a significant performance discrepancy between MAPG methods and state-of-the-art multi-agent value-based approaches. In this paper, we investigate causes that hinder the performance of MAPG algorithms and present a multi-agent decomposed policy gradient method (DOP). This method introduces the idea of value function decomposition into the multi-agent actor-critic framework. Based on this idea, DOP supports efficient off-policy learning and addresses the issue of *centralized-decentralized mismatch* and credit assignment in both discrete and continuous action spaces. We formally show that DOP critics have sufficient representational capability to guarantee convergence. In addition, empirical evaluations on the StarCraft II micromanagement benchmark and multi-agent particle environments demonstrate that DOP outperforms both state-of-the-art value-based and policy-based multi-agent reinforcement learning algorithms. Demonstrative videos are available at *https://sites.google.com/view/dop-mapg/*.

## 1 Introduction

Cooperative multi-agent reinforcement learning (MARL) has achieved great progress in recent years (Hughes et al., 2018; Jaques et al., 2019; Vinyals et al., 2019; Zhang et al., 2019; Baker et al., 2020; Wang et al., 2020c). Advances in valued-based MARL (Sunehag et al., 2018; Rashid et al., 2018; Son et al., 2019; Wang et al., 2020e) contribute significantly to the progress, achieving state-of-the-art performance on challenging tasks, such as StarCraft II micromanagement (Samvelyan et al., 2019). However, these value-based methods present a major challenge for stability and convergence in multi-agent settings (Wang et al., 2020a), which is further exacerbated in continuous action spaces. Policy gradient methods hold great promise to resolve these challenges. MADDPG (Lowe et al., 2017) and COMA (Foerster et al., 2018) are two representative methods that adopt the paradigm of centralized critic with decentralized actors (CCDA), which not only deals with the issue of non-stationarity (Foerster et al., 2017; Hernandez-Leal et al., 2017) by conditioning the centralized critic on global history and actions but also maintains scalable decentralized execution via conditioning policies on local history. Several subsequent works make improvements to the CCDA framework by introducing the mechanism of recursive reasoning (Wen et al., 2019) or attention (Iqbal & Sha, 2019).

Despite the progress, most of the multi-agent policy gradient (MAPG) methods do not provide satisfying performance, e.g., significantly underperforming value-based methods on benchmark tasks (Samvelyan et al., 2019). In this paper, we analyze this discrepancy and pinpoint three major issues that hinder the performance of MAPG methods. (1) Current stochastic MAPG methods do not support off-policy learning, partly because using common off-policy learning techniques is computationally expensive in multi-agent settings. (2) In the CCDA paradigm, the suboptimality of one agent's policy can propagate through the centralized joint critic and negatively affect policy learning of other agents, causing catastrophic miscoordination, which we call *centralized-decentralized mismatch* (CDM). (3) For deterministic MAPG methods, realizing efficient credit assignment (Tumer et al., 2002; Agogino & Tumer, 2004) with a single global reward signal largely remains challenging.

---

[*]Equal Contribution. Listing order is random.

In this paper, we find that these problems can be addressed by introducing the idea of value decomposition into the multi-agent actor-critic framework and learning a centralized but factorized critic. This framework decomposes the centralized critic as a weighted linear summation of individual critics that condition on local actions. This decomposition structure not only enables scalable learning on the critic, but also brings several benefits. It enables tractable off-policy evaluations of stochastic policies, attenuates the CDM issues, and also implicitly learns an efficient multi-agent credit assignment. Based on this decomposition, we develop efficient off-policy multi-agent decomposed policy gradient methods for both discrete and continuous action spaces.

A drawback of an linearly decomposed critic is its limited representational capacity (Wang et al., 2020b), which may induce bias in value estimations. However, we show that this bias does not violate the policy improvement guarantee of policy gradient methods and that using decomposed critics can largely reduce the variance in policy updates. In this way, a decomposed critic achieves a great bias-variance trade-off.

We evaluate our methods on both the StarCraft II micromanagement benchmark (Samvelyan et al., 2019) (discrete action spaces) and multi-agent particle environments (Lowe et al., 2017; Mordatch & Abbeel, 2018) (continuous action spaces). Empirical results show that DOP is very stable across different runs and outperforms other MAPG algorithms by a wide margin. Moreover, to our best knowledge, stochastic DOP provides the first MAPG method that outperforms state-of-the-art valued-based methods in discrete-action benchmark tasks.

**Related works on value decomposition methods.** In value-based MARL, value decomposition (Guestrin et al., 2002b; Castellini et al., 2019) is widely used. These methods learn local Q-value functions for each agent, which are combined with a learnable mixing function to produce global action values. In VDN (Sunehag et al., 2018), the mixing function is an arithmetic summation. QMIX (Rashid et al., 2018; 2020) proposes a non-linear monotonic factorization structure. QTRAN (Son et al., 2019) and QPLEX (Wang et al., 2020b) further extend the class of value functions that can be represented. NDQ (Wang et al., 2020e) addresses the miscoordination problem by learning nearly decomposable architectures. A concurrent work (de Witt et al., 2020) finds that a decomposed centralized critic in QMIX style can improve the performance of MADDPG for learning in continuous action spaces. In this paper, we study how and why linear value decomposition can enable efficient policy-based learning in both discrete and continuous action spaces. In Appendix F, we discuss how DOP is related to recent progress in multi-agent reinforcement learning and provide detailed comparisons with existing multi-agent policy gradient methods.

## 2 BACKGROUND

We consider fully cooperative multi-agent tasks that can be modelled as a Dec-POMDP (Oliehoek et al., 2016) $G=\langle I, S, A, P, R, \Omega, O, n, \gamma \rangle$, where $I$ is the finite set of agents, $\gamma \in [0, 1)$ is the discount factor, and $s \in S$ is the true state of the environment. At each timestep, each agent $i$ receives an observation $o_i \in \Omega$ drawn according to the observation function $O(s, i)$ and selects an action $a_i \in A$, forming a joint action $\boldsymbol{a} \in A^n$, leading to a next state $s'$ according to the transition function $P(s'|s, \boldsymbol{a})$ and a reward $r = R(s, \boldsymbol{a})$ shared by all agents. Each agent learns a policy $\pi_i(a_i|\tau_i; \theta_i)$, which is parameterized by $\theta_i$ and conditioned on the local history $\tau_i \in T \equiv (\Omega \times A)^*$. The joint policy $\boldsymbol{\pi}$, with parameters $\theta = \langle \theta_1, \cdots, \theta_n \rangle$, induces a joint action-value function: $Q_{tot}^{\boldsymbol{\pi}}(\boldsymbol{\tau}, \boldsymbol{a}) = \mathbb{E}_{s_{0:\infty}, \boldsymbol{a}_{0:\infty}}[\sum_{t=0}^{\infty} \gamma^t R(s_t, \boldsymbol{a}_t) | s_0 = s, \boldsymbol{a}_0 = \boldsymbol{a}, \boldsymbol{\pi}]$. We consider both discrete and continuous action spaces, for which stochastic and deterministic policies are learned, respectively. To distinguish deterministic policies, we denote them by $\boldsymbol{\mu} = \langle \mu_1, \cdots, \mu_n \rangle$.

**Multi-Agent Policy Gradients** The *centralized training with decentralized execution* (CTDE) paradigm (Foerster et al., 2016; Wang et al., 2020d) has recently attracted attention for its ability to address non-stationarity while maintaining decentralized execution. Learning a centralized critic with decentralized actors (CCDA) is an efficient approach that exploits the CTDE paradigm. MADDPG and COMA are two representative examples. MADDPG (Lowe et al., 2017) learns deterministic policies in continuous action spaces and uses the following gradients to update policies:

$$g = \mathbb{E}_{\boldsymbol{\tau}, \boldsymbol{a} \sim \mathcal{D}} \left[ \sum_i \nabla_{\theta_i} \mu_i(\tau_i) \nabla_{a_i} Q_{tot}^{\boldsymbol{\mu}}(\boldsymbol{\tau}, \boldsymbol{a})|_{a_i = \mu_i(\tau_i)} \right], \tag{1}$$

and $\mathcal{D}$ is a replay buffer. COMA (Foerster et al., 2018) updates stochastic policies using the gradients:

$$g = \mathbb{E}_{\boldsymbol{\pi}} \left[ \sum_i \nabla_{\theta_i} \log \pi_i(a_i|\tau_i) A_i^{\boldsymbol{\pi}}(\boldsymbol{\tau}, \boldsymbol{a}) \right], \tag{2}$$

where $A_i^{\boldsymbol{\pi}}(\boldsymbol{\tau}, \boldsymbol{a}) = Q_{tot}^{\boldsymbol{\pi}}(\boldsymbol{\tau}, \boldsymbol{a}) - \sum_{a_i'} Q_{tot}^{\boldsymbol{\pi}}(\boldsymbol{\tau}, (\boldsymbol{a}_{-i}, a_i'))$ is a counterfactual advantage ($\boldsymbol{a}_{-i}$ is the joint action other than agent $i$) that deals with the issue of credit assignment and reduces variance.

## 3 ANALYSIS

In this section, we investigate challenges that limit the performance of state-of-the-art multi-agent policy gradient methods.

### 3.1 OFF-POLICY LEARNING FOR MULTI-AGENT STOCHASTIC POLICY GRADIENTS

Efficient stochastic policy learning in single-agent settings relies heavily on using off-policy data (Lillicrap et al., 2015; Wang et al., 2016; Fujimoto et al., 2018; Haarnoja et al., 2018), which is not supported by existing stochastic MAPG methods (Foerster et al., 2018). In the CCDA framework, off-policy policy evaluation—estimating $Q_{tot}^{\boldsymbol{\pi}}$ from data drawn from behavior policies $\boldsymbol{\beta} = \langle \beta_1, \ldots, \beta_n \rangle$—encounters major challenges. Importance sampling (Meuleau et al., 2000; Jie & Abbeel, 2010; Levine & Koltun, 2013) is a simple way to correct for the discrepancy between $\boldsymbol{\pi}$ and $\boldsymbol{\beta}$, but, it requires computing $\prod_i \frac{\pi_i(a_i|\tau_i)}{\beta_i(a_i|\tau_i)}$, whose variance grows exponentially with the number of agents in multi-agent settings. An alternative is to extend the tree backup technique (Precup et al., 2000; Munos et al., 2016) to multi-agent settings and use the $k$-step tree backup update target for training the critic:

$$y^{TB} = Q_{tot}^{\boldsymbol{\pi}}(\boldsymbol{\tau}, \boldsymbol{a}) + \sum_{t=0}^{k-1} \gamma^t \left( \prod_{l=1}^t \lambda \boldsymbol{\pi}(\boldsymbol{a}_l|\boldsymbol{\tau}_l) \right) [r_t + \gamma \mathbb{E}_{\boldsymbol{\pi}}[Q_{tot}^{\boldsymbol{\pi}}(\boldsymbol{\tau}_{t+1}, \cdot)] - Q_{tot}^{\boldsymbol{\pi}}(\boldsymbol{\tau}_t, \boldsymbol{a}_t)], \tag{3}$$

where $\boldsymbol{\tau}_0 = \boldsymbol{\tau}$, $\boldsymbol{a}_0 = \boldsymbol{a}$. However, the complexity of computing $\mathbb{E}_{\boldsymbol{\pi}}[Q_{tot}^{\boldsymbol{\pi}}(\boldsymbol{\tau}_{t+1}, \cdot)]$ is $O(|A|^n)$, which becomes intractable when the number of agents is large. Therefore, it is challenging to develop off-policy stochastic MAPG methods.

### 3.2 THE CENTRALIZED-DECENTRALIZED MISMATCH ISSUE

In the centralized critic with decentralized actors (CCDA) framework, agents learn individual policies, $\pi_i(a_i|\tau_i; \theta_i)$, conditioned on the local observation-action history. However, the gradients for updating these policies are dependent on the centralized joint critic, $Q_{tot}^{\boldsymbol{\pi}}(\boldsymbol{\tau}, \boldsymbol{a})$ (see Eq. 1 and 2), which introduces the influence of actions of other agents. Intuitively, gradient updates will move an agent in the direction that can increase the global Q value, but the presence of other agents' actions incurs large variance in the estimates of such directions.

Formally, the variance of policy gradients for agent $i$ at $(\tau_i, a_i)$ is dependent on other agents' actions:

$$\begin{aligned} &\text{Var}_{\boldsymbol{a}_{-i} \sim \boldsymbol{\pi}_{-i}} \left[ Q_{tot}^{\boldsymbol{\pi}}(\boldsymbol{\tau}, (a_i, \boldsymbol{a}_{-i})) \nabla_{\theta_i} \log \pi_i(a_i|\tau_i) \right] \\ =&\text{Var}_{\boldsymbol{a}_{-i} \sim \boldsymbol{\pi}_{-i}} \left[ Q_{tot}^{\boldsymbol{\pi}}(\boldsymbol{\tau}, (a_i, \boldsymbol{a}_{-i})) \right] \left( \nabla_{\theta_i} \log \pi_i(a_i|\tau_i) \right) \left( \nabla_{\theta_i} \log \pi_i(a_i|\tau_i) \right)^{\text{T}}, \end{aligned} \tag{4}$$

where $\text{Var}_{\boldsymbol{a}_{-i}}[Q_{tot}^{\boldsymbol{\pi}}(\boldsymbol{\tau}, (a_i, \boldsymbol{a}_{-i}))]$ can be very large due to the exploration or suboptimality of other agents' policies, which may cause suboptimality in individual policies. For example, suppose that the optimal joint action under $\boldsymbol{\tau}$ is $\boldsymbol{a}^* = \langle a_1^*, \ldots, a_n^* \rangle$. When $\mathbb{E}_{\boldsymbol{a}_{-i} \sim \boldsymbol{\pi}_{-i}}[Q_{tot}^{\boldsymbol{\pi}}(\boldsymbol{\tau}, (a_i^*, \boldsymbol{a}_{-i}))] < 0$, $\pi_i(a_i^*|\tau_i)$ will decrease, possibly resulting in a suboptimal $\pi_i$. This becomes problematic because a negative feedback loop is created, in which the joint critic is affected by the suboptimality of agent $i$, which disturbs policy updates of other agents. We call this issue *centralized-decentralized mismatch* (CDM).

**Does CDM occur in practice for state-of-the-art algorithms?** To answer this question, we carry out a case study in Sec. 5.1. We can see that the variance of DOP gradients is significantly smaller than COMA and MADDPG (Fig. 2 left). This smaller variance enables DOP to outperform other algorithms (Fig. 2 middle). We will explain this didactic example in detail in Sec. 5.1. In Sec. 5.2 and 5.3, we further show that CDM is exacerbated in sequential decision-making settings, causing divergence even after a near-optimal strategy has been learned.

### 3.3 Credit Assignment for Multi-Agent Deterministic Policy Gradients

MADDPG (Lowe et al., 2017) and MAAC (Iqbal & Sha, 2019) extend deterministic policy gradient algorithms (Silver et al., 2014; Lillicrap et al., 2015) to multi-agent settings, enabling efficient off-policy learning in continuous action spaces. However, they leave the issue of credit assignment (Tumer et al., 2002; Agogino & Tumer, 2004) largely untouched in fully cooperative settings, where agents learn policies from a single global reward signal. In stochastic cases, COMA assigns credits by designing a counterfactual baseline (Eq. 2). However, it is not straightforward to extend COMA to deterministic policies, since the output of polices is no longer a probability distribution. As a result, it remains challenging to realize efficient credit assignment in deterministic cases.

## 4 Decomposed Off-Policy Policy Gradients

To address the limitations of existing MAPG methods discussed in Sec. 3, we introduce the idea of value decomposition into the multi-agent actor-critic framework and propose a *Decomposed Off-Policy policy gradient* (DOP) method. We factor the centralized critic as a weighted summation of individual critics across agents:

$$Q_{tot}^{\phi}(\boldsymbol{\tau}, \mathbf{a}) = \sum_i k_i(\boldsymbol{\tau}) Q_i^{\phi_i}(\boldsymbol{\tau}, a_i) + b(\boldsymbol{\tau}), \tag{5}$$

where $\phi$ and $\phi_i$ are parameters of the global and local Q functions, respectively, and $k_i \geq 0$ and $b$ are generated by learnable networks whose inputs are global observation-action histories. In the following sections, we show that this linear decomposition helps address existing limitations of previous methods. A concern is the limited expressivity of linear decomposition (Wang et al., 2020b), which may introduce bias in value estimations. We will show that this limitation does not violate the policy improvement guarantee of DOP.

Fig. 1 shows the architecture for learning decomposed critics. We learn individual critics $Q_i^{\phi_i}$ by backpropagating gradients from global TD updates dependent on the joint global reward, i.e., $Q_i^{\phi_i}$ is learned implicitly rather than from any reward specific to agent $i$. We enforce $k_i \geq 0$ by applying an absolute activation function at the last layer of the network. The network structure is described in detail in Appendix H.

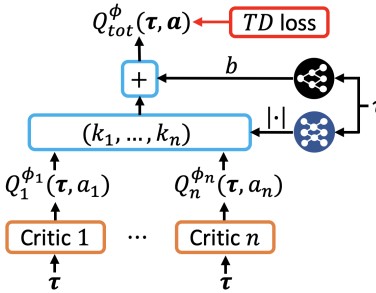

Based on the critic decomposition learning, the following sections will introduce decomposed off-policy policy gradients for learning stochastic policies and deterministic policies, respectively. Similar to other actor-critic methods, DOP alternates between *policy evaluation*—estimating the value function for a policy—and *policy improvement*—using the value function to update the policy (Barto et al., 1983).

Figure 1: A Decomposed critic.

### 4.1 Stochastic Decomposed Off-Policy Policy Gradients

For learning stochastic policies, the linearly decomposed critic plays an essential role in enabling tractable multi-agent tree backup for off-policy policy evaluation and attenuating the CDM issue while maintaining provable policy improvement.

#### 4.1.1 Off-Policy Learning

**Policy Evaluation: Train the Critic** As discussed in Sec. 3.1, using tree backup (Eq. 3) to carry out multi-agent off-policy policy evaluation requires calculating $\mathbb{E}_{\boldsymbol{\pi}}[Q_{tot}^{\phi}(\boldsymbol{\tau}_{t+1}, \cdot)]$, which needs $O(|A|^n)$ steps of summation when a joint critic is used. Fortunately, using the linearly decomposed critic, DOP reduces the complexity of computing this expectation to $O(n|A|)$:

$$\mathbb{E}_{\boldsymbol{\pi}}[Q_{tot}^{\phi}(\boldsymbol{\tau}, \cdot)] = \sum_i k_i(\boldsymbol{\tau}) \mathbb{E}_{\pi_i}[Q_i^{\phi_i}(\boldsymbol{\tau}, \cdot)] + b(\boldsymbol{\tau}), \tag{6}$$

making the tree backup technique tractable (detailed proof can be found in Appendix A.1). Another challenge of using multi-agent tree backup (Eq. 3) is that the coefficient $c_t = \prod_{l=1}^t \lambda \boldsymbol{\pi}(\boldsymbol{a}_l|\boldsymbol{\tau}_l)$ decays

as $t$ gets larger, which may lead to relatively lower training efficiency. To solve this issue, we propose to mix off-policy tree backup updates with on-policy $TD(\lambda)$ updates to trade off sample efficiency and training efficiency. Formally, DOP minimizes the following loss for training the critic:

$$\mathcal{L}(\phi) = \kappa \mathcal{L}_{\boldsymbol{\beta}}^{\text{DOP-TB}}(\phi) + (1 - \kappa)\mathcal{L}_{\boldsymbol{\pi}}^{\text{On}}(\phi) \tag{7}$$

where $\kappa$ is a scaling factor, $\boldsymbol{\beta}$ is the joint behavior policy, and $\phi$ is the parameters of the critic. The first loss item is $\mathcal{L}_{\boldsymbol{\beta}}^{\text{DOP-TB}}(\phi) = \mathbb{E}_{\boldsymbol{\beta}}[(y^{\text{DOP-TB}} - Q_{tot}^{\phi}(\boldsymbol{\tau}, \boldsymbol{a}))^2]$, where $y^{\text{DOP-TB}}$ is the update target of the proposed $k$-step decomposed multi-agent tree backup algorithm:

$$y^{\text{DOP-TB}} = Q_{tot}^{\phi'}(\boldsymbol{\tau}, \boldsymbol{a}) + \sum_{t=0}^{k-1} \gamma^t c_t \left[ r_t + \gamma \sum_i k_i(\boldsymbol{\tau}_{t+1})\mathbb{E}_{\pi_i}[Q_i^{\phi'_i}(\boldsymbol{\tau}_{t+1}, \cdot)] + b(\boldsymbol{\tau}_{t+1}) - Q_{tot}^{\phi'}(\boldsymbol{\tau}_t, \boldsymbol{a}_t) \right]. \tag{8}$$

Here, $\phi'$ is the parameters of a target critic, and $\boldsymbol{a}_t \sim \boldsymbol{\beta}(\cdot|\boldsymbol{\tau}_t)$. The second loss item is $\mathcal{L}_{\boldsymbol{\pi}}^{\text{On}}(\phi) = \mathbb{E}_{\boldsymbol{\pi}}[(y^{\text{On}} - Q_{tot}^{\phi}(\boldsymbol{\tau}, \boldsymbol{a}))^2]$, where $y^{\text{On}}$ is the on-policy update target as in $TD(\lambda)$:

$$y^{\text{On}} = Q_{tot}^{\phi'}(\boldsymbol{\tau}, \boldsymbol{a}) + \sum_{t=0}^{\infty} (\gamma\lambda)^t \left[ r_t + \gamma Q_{tot}^{\phi'}(\boldsymbol{\tau}_{t+1}, \boldsymbol{a}_{t+1}) - Q_{tot}^{\phi'}(\boldsymbol{\tau}_t, \boldsymbol{a}_t) \right]. \tag{9}$$

In practice, we use two buffers, an on-policy buffer for computing $\mathcal{L}_{\boldsymbol{\pi}}^{\text{On}}(\phi)$ and an off-policy buffer for estimating $\mathcal{L}_{\boldsymbol{\beta}}^{\text{DOP-TB}}(\phi)$.

**Policy Improvement: Train Actors** Using the linearly decomposed critic architecture, we can derive the following on-policy policy gradients for learning stochastic policies:

$$g = \mathbb{E}_{\boldsymbol{\pi}} \left[ \sum_i k_i(\boldsymbol{\tau})\nabla_{\theta_i} \log \pi_i(a_i|\tau_i; \theta_i)Q_i^{\phi_i}(\boldsymbol{\tau}, a_i) \right] \tag{10}$$

In Appendix A.2, we provide the detailed derivation and an off-policy version of stochastic policy gradients. This update rule reveals two important insights. (1) With a linearly decomposed critic, each agent's policy update only depends on the individual critic $Q_i^{\phi_i}$. (2) Learning the decomposed critic implicitly realizes multi-agent credit assignment, because the individual critic provides credit information for each agent to improve its policy in the direction of increasing the global expected return. Moreover, Eq. 10 is also the policy gradients when assigning credits via the aristocrat utility (Wolpert & Tumer, 2002) (Appendix A.2). Eq. 7 and 10 form the core of our DOP algorithm for learning stochastic policies, which we call *stochastic DOP* and is described in detail in Appendix E.

**The CDM Issue** occurs when decentralized policies' suboptimality exacerbates each other through the joint critic. As an agent's stochastic DOP gradients do not rely on the actions of other agents, they attenuate the effect of CDM. We empirically show that DOP can reduce variance in policy gradients in Sec. 5.1 and can attenuate the CDM issue in complex tasks in Sec. 5.2.1.

### 4.1.2 STOCHASTIC DOP POLICY IMPROVEMENT THEOREM

In this section, we theoretically demonstrate that stochastic DOP can converge to local optimal despite the fact that a linearly decomposed critic has limited representational capability. Since an accurate analysis for a complex function approximator (e.g., neural network) is difficult, we adopt several mild assumptions used in previous work (Feinberg et al., 2018; Degris et al., 2012).

We first show that the linearly decomposed structure ensures that the learned local value functions $Q_i^{\phi_i}(\boldsymbol{\tau}, a_i)$ preserve the order of $Q_i^{\boldsymbol{\pi}}(\boldsymbol{\tau}, a_i) = \sum_{\boldsymbol{a}_{-i}} \boldsymbol{\pi}_{-i}(\boldsymbol{a}_{-i}|\tau_{-i})Q_{tot}^{\boldsymbol{\pi}}(\boldsymbol{\tau}, \boldsymbol{a})$ for a wide range of function class.

**Fact 1.** *Under mild assumptions, when value evaluation converges, $\forall \boldsymbol{\pi}$, $Q_i^{\phi_i}$ satisfies that*

$$Q_i^{\boldsymbol{\pi}}(\boldsymbol{\tau}, a_i) \geq Q_i^{\boldsymbol{\pi}}(\boldsymbol{\tau}, a'_i) \iff Q_i^{\phi_i}(\boldsymbol{\tau}, a_i) \geq Q_i^{\phi_i}(\boldsymbol{\tau}, a'_i), \quad \forall \boldsymbol{\tau}, a_i, a'_i.$$

Detailed proof of Fact 1 can be found in Appendix C.1 as well as more detailed discussion of its implications. Furthermore, we prove the following proposition to show that policy improvement can be guaranteed as long as the function class expressed by $Q_i^{\phi_i}$ is sufficiently large and the loss of critic training is minimized.

**Proposition 1.** *Suppose the function class expressed by $Q_i^{\phi_i}(\boldsymbol{\tau}, a_i)$ is sufficiently large (e.g. neural networks) and the following loss $L(\phi)$ is minimized*

$$L(\phi) = \sum_{\boldsymbol{a}, \boldsymbol{\tau}} p(\boldsymbol{\tau}) \boldsymbol{\pi}(\boldsymbol{a}|\boldsymbol{\tau}) (Q_{tot}^{\boldsymbol{\pi}}(\boldsymbol{\tau}, \boldsymbol{a}) - Q_{tot}^{\phi}(\boldsymbol{\tau}, \boldsymbol{a}))^2,$$

*where $Q_{tot}^{\phi}(\boldsymbol{\tau}, \mathbf{a}) \equiv \sum_i k_i(\boldsymbol{\tau}) Q_i^{\phi_i}(\boldsymbol{\tau}, a_i) + b(\boldsymbol{\tau})$. Then, we have*

$$g = \mathbb{E}_{\boldsymbol{\pi}} \left[ \sum_i \nabla_{\theta_i} \log \pi_i(a_i|\tau_i; \theta_i) Q^{\boldsymbol{\pi}}(\boldsymbol{\tau}, \boldsymbol{a}) \right]$$

$$= \mathbb{E}_{\boldsymbol{\pi}} \left[ \sum_i k_i(\boldsymbol{\tau}) \nabla_{\theta_i} \log \pi_i(a_i|\tau_i; \theta_i) Q_i^{\phi_i}(\boldsymbol{\tau}, a_i) \right],$$

*which means stochastic DOP policy gradients are the same as those calculated using centralized critics (Eq. 2). Therefore, policy improvement is guaranteed.*

The proof can be found in Appendix C.2, which is inspired by Wang et al. (2020a).

## 4.2 DETERMINISTIC DECOMPOSED OFF-POLICY POLICY GRADIENTS

### 4.2.1 OFF-POLICY LEARNING

To enable efficient learning with continuous actions, we propose *deterministic DOP*. As in single-agent settings, because deterministic policy gradient methods avoid the integral over actions, it greatly eases the cost of off-policy learning (Silver et al., 2014). For **policy evaluation**, we train the critic by minimizing the following TD loss:

$$\mathcal{L}(\phi) = \mathbb{E}_{(\boldsymbol{\tau}_t, r_t, \boldsymbol{a}_t, \boldsymbol{\tau}_{t+1}) \sim \mathcal{D}} \left[ \left( r_t + \gamma Q_{tot}^{\phi'}(\boldsymbol{\tau}_{t+1}, \boldsymbol{\mu}(\boldsymbol{\tau}_{t+1}; \theta')) - Q_{tot}^{\phi}(\boldsymbol{\tau}_t, \boldsymbol{a}_t) \right)^2 \right], \quad (11)$$

where $\mathcal{D}$ is a replay buffer, and $\phi'$, $\theta'$ are the parameters of the target critic and actors, respectively. For **policy improvement**, we derive the following deterministic DOP policy gradients:

$$g = \mathbb{E}_{\boldsymbol{\tau} \sim \mathcal{D}} \left[ \sum_i k_i(\boldsymbol{\tau}) \nabla_{\theta_i} \mu_i(\tau_i; \theta_i) \nabla_{a_i} Q_i^{\phi_i}(\boldsymbol{\tau}, a_i)|_{a_i = \mu_i(\tau_i; \theta_i)} \right]. \quad (12)$$

Detailed proof can be found in Appendix B.1. Similar to the stochastic case, This result reveals that updates of individual deterministic policies depend on local critics when a linearly decomposed critic is used. Based on Eq. 11 and Eq. 12, we develop the DOP algorithm for learning deterministic policies in continuous action spaces, which is described in Appendix E and called *deterministic DOP*.

### 4.2.2 REPRESENTATION CAPACITY OF DETERMINISTIC DOP CRITICS

In continuous and smooth environments, we first show that a DOP critic has sufficient expressive capability to represent Q values in the proximity of $\boldsymbol{\mu}(\boldsymbol{\tau}), \forall \boldsymbol{\tau}$ with a bounded error. For simplicity, we denote $O_\delta(\boldsymbol{\tau}) = \{\boldsymbol{a}| \parallel \boldsymbol{a} - \boldsymbol{\mu}(\boldsymbol{\tau}) \parallel_2 \le \delta\}$.

**Fact 2.** *Assume that $\forall \boldsymbol{\tau}, \boldsymbol{a}, \boldsymbol{a}' \in O_\delta(\boldsymbol{\tau}), \parallel \nabla_{\boldsymbol{a}} Q_{tot}^{\boldsymbol{\mu}}(\boldsymbol{\tau}, \boldsymbol{a}) - \nabla_{\boldsymbol{a}'} Q_{tot}^{\boldsymbol{\mu}}(\boldsymbol{\tau}, \boldsymbol{a}') \parallel_2 \le L \parallel \boldsymbol{a} - \boldsymbol{a}' \parallel_2$. The estimation error of a DOP critic can be bounded by $O(L\delta^2)$ for $\boldsymbol{a} \in O_\delta(\boldsymbol{\tau}), \forall \boldsymbol{\tau}$.*

Detailed proof can be found in Appendix D. Here we assume that the gradients of Q-values with respect to actions are Lipschitz smooth under the deterministic policy $\boldsymbol{\mu}$. This assumption is reasonable given that Q-values of most continuous environments with continuous policies are rather smooth.

We further show that when Q-values in the proximity of $\boldsymbol{\mu}(\boldsymbol{\tau}), \forall \boldsymbol{\tau}$ are well estimated with a bounded error, deterministic DOP policy gradients are good approximation to the true gradients (Eq. 1). Approximately, $|\nabla_{a_i} Q_{tot}^{\boldsymbol{\mu}}(\boldsymbol{\tau}, \boldsymbol{a}) - \nabla_{a_i} k_i(\boldsymbol{\tau}) \nabla_{a_i} Q_i^{\phi_i}(\boldsymbol{\tau}, a_i)| \sim O(L\delta), \forall i$ when $\delta \ll 1$. For detailed proof, we refer readers to Appendix D.

## 5 EXPERIMENTS

We design experiments to answer the following questions: (1) Does the CDM issue commonly exist and can decomposed critics attenuate it? (Sec. 5.1, 5.2.1, and 5.3) (2) Can our decomposed multi-agent tree backup algorithm improve the efficiency of off-policy learning? (Sec. 5.2.1) (3) Can deterministic DOP learn reasonable credit assignment? (Sec. 5.3) (4) Can DOP outperform state-of-the-art MARL algorithms? For evaluation, all the results are averaged over 12 different random seeds and are shown with $95\%$ confidence intervals.

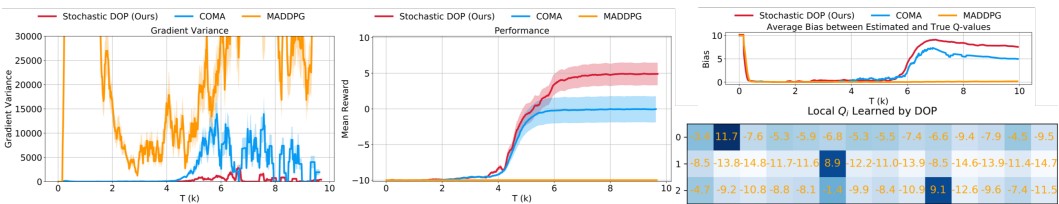

Figure 2: Bias-variance trade-off of DOP on the didactic example. Left: gradient variance; Middle: Performance; Right: Average bias in Q estimations; Right-bottom: the element in $i$th row and $j$th column is the local Q value learned by DOP for agent $i$ taking action $j$.

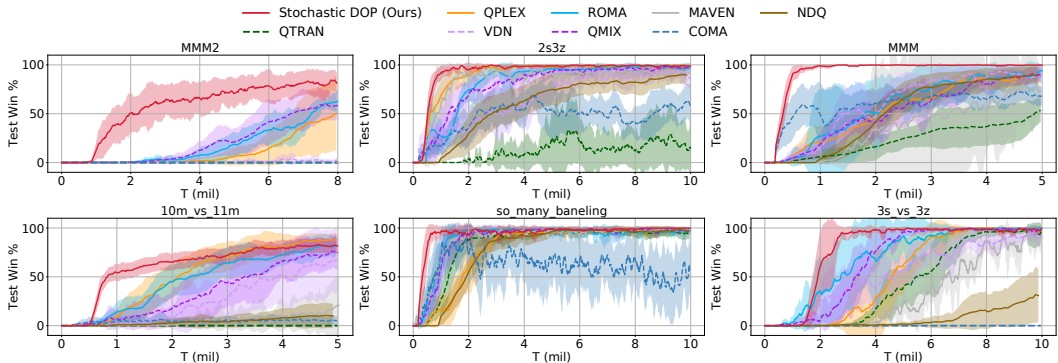

Figure 3: Comparisons with baselines on the SMAC benchmark.

## 5.1 DIDACTIC EXAMPLE: THE CDM ISSUE AND BIAS-VARIANCE TRADE-OFF

We use a didactic example to demonstrate how DOP attenuates CDM and achieves bias-variance trade-off. In a state-less game with 3 agents and 14 actions, if agents take action1, 5, 9, respectively, they get a team reward of 10; otherwise −10. We train stochastic DOP, COMA, and MADDPG for $10K$ timesteps and show the gradient variance, value estimation bias, and learning curves in Fig. 2. Gumbel-Softmax trick (Jang et al., 2017; Maddison et al., 2017) is used to enable MADDPG to learn in discrete action spaces.

Fig. 2-right shows the average bias in the estimations of all Q values. We see that linear decomposition introduces extra estimation errors. However, the variance of DOP policy gradients is much smaller than other algorithms (Fig. 2-left). As discussed in Sec. 3.2, large variance of other algorithms is due to the CDM issue that undecomposed joint critics are affected by actions of all agents. Free from the influence of other agents, DOP preserves the order of local Q-values (bottom of Fig. 2-right) and effectively reduces the variance of policy gradients. In this way, DOP sacrifices value estimation accuracy for accurate and low-variance policy gradients, which explains why it can outperform other algorithms (Fig. 2-middle).

## 5.2 DISCRETE ACTION SPACES: THE STARCRAFT II MICROMANAGEMENT BENCHMARK

We evaluate stochastic DOP on the challenging SMAC benchmark (Samvelyan et al., 2019) for its high control complexity. We compare our method with the state-of-the-art multi-agent stochastic policy gradient method (COMA), value-based methods (VDN, QMIX, QTRAN (Son et al., 2019), NDQ (Wang et al., 2020e), and QPLEX (Wang et al., 2020b)), exploration method (MAVEN, Mahajan et al. (2019)), and role-based method (ROMA, Wang et al. (2020c)). For stochastic DOP, we fix the hyperparameter setting and network structure in all experiments which are described in Appendix H. For baselines, we use their default hyperparameter settings that have been fine-tuned on the SMAC benchmark. Results are shown in Fig. 3. Stochastic DOP significantly outperforms all the baselines by a wide margin. To our best knowledge, this is the first time that a MAPG method has significantly better performance than state-of-the-art value-based methods.

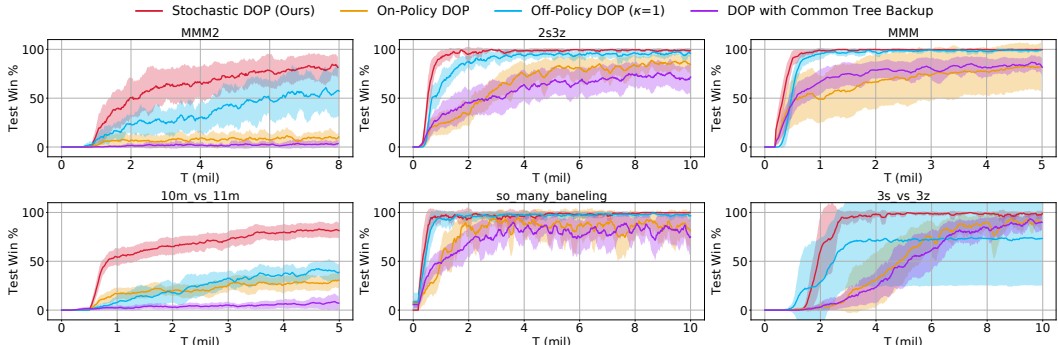

Figure 4: Comparisons with ablations on the SMAC benchmark.

### 5.2.1 ABLATIONS

Stochastic DOP has three main components: (a) off-policy policy evaluations, (b) the decomposed critic, and (c) decomposed multi-agent tree backup. By design, component (a) improves sample efficiency, component (b) can attenuate the CDM issue, and component (c) makes off-policy policy evaluations tractable. We test the contribution of each component by carrying out the following ablation studies.

**Off-Policy Learning** In our method, $\kappa$ controls the "off-policyness" of training. For DOP, we set $\kappa$ to $0.5$. To demonstrate the effect of off-policy learning, we change $\kappa$ to $0$ and $1$ and compare the performance. In Fig. 4, we can see that both DOP and off-policy DOP perform much better than the on-policy version ($\kappa$=0), highlighting the importance of using off-policy data. Moreover, purely off-policy learning generally needs more samples to achieve similar performance to DOP. Mixing with on-policy data can largely improve training efficiency.

**The CDM Issue** *On-Policy DOP* uses the same decomposed critic structure as DOP, but is trained only with on-policy data and does not use tree backup. The only difference between *On-Policy DOP* and COMA is that the former one uses a decomposed joint critic. Therefore, given that a COMA critic has a more powerful expression capacity than a DOP critic, the outperformance of *On-Policy DOP* against COMA shows the effect of CDM. COMA is not stable and may diverge after a near-optimal policy has been learned. For example, on map `so_many_baneling`, COMA policies degenerate after 2M steps. In contrast, On-Policy DOP can converge with efficiency and stability.

**Decomposed Multi-Agent Tree Backup** *DOP with Common Tree Backup* (DOP without component (c)) is the same as DOP except that $\mathbb{E}_{\pi}[Q_{tot}^{\phi}(\tau, \cdot)]$ is estimated by sampling 200 joint actions from $\pi$. Here, we estimate this expectation by sampling because direct computation is intractable (for example, $20^{10}$ summations are needed on the map `MMM`). Fig. 4 shows that when the number of agents increases, sampling becomes less efficient, and common tree backup performs even worse than *On-Policy DOP*. In contrast, DOP with decomposed tree backup can quickly and stably converge using a similar number of summations.

### 5.3 CONTINUOUS ACTION SPACES: MULTI-AGENT PARTICLE ENVIRONMENTS

We evaluate deterministic DOP on multi-agent particle environments (MPE, (Mordatch & Abbeel, 2018)), where agents take continuous actions in continuous spaces. We compare our method with MADDPG (Lowe et al., 2017) and MAAC (Iqbal & Sha, 2019). Hyperparameters and the network structure are fixed for deterministic DOP across experiments, which are described in Appendix H.

**The CDM Issue** We use task `Aggregation` as an example to show that deterministic DOP attenuates the CDM issue. In this task, $5$ agents navigate to one landmark. Only when all agents reach the landmark will they get a team reward of $10$ and successfully end the episode; otherwise, an episode ends after 25 timesteps and agents get a reward of $-10$. `Aggregation` is a typical example where other agents' actions can influence an agent's local policy through an undecomposed joint critic. Intuitively, as long as one agent does not reach the landmark, the centralized Q value is negative, confusing other agents who get to the landmark. This intuition is supported by the empirical results

Figure 5: Left and middle: performance comparisons with COMA and MAAC on MPE. Right: The learned credit assignment mechanism on task `Mill` by deterministic DOP.

shown in Fig. 5-left – methods with undecomposed critics can find rewarding configurations but then quickly diverge, while DOP converges with stability.

**Credit Assignment** We use task `Mill` to show that DOP can learn effective credit assignment mechanisms. In this task, 10 agents need to rotate a millstone clockwise. They can push the millstone clockwise or counterclockwise with force between $0$ and $1$. If the millstone's angular velocity, $\omega$, gets greater than 30, agents are rewarded 3 per step. If $\omega$ exceeds 100 in 10 steps, the agents win the episode and get a reward of 10; otherwise, they lose and get a punishment of -10. Fig. 5-right shows that deterministic DOP can gradually learn a reasonable credit assignment during training, where rotating the millstone clockwise has much larger Q-values. This explains why deterministic DOP outperforms previous state-of-the-art deterministic MAPG methods, as shown in Fig. 5-middle.

## 6 CLOSING REMARKS

This paper pinpointed drawbacks that hinder the performance of state-of-the-art MAPG algorithms: on-policy learning of stochastic policy gradient methods, the centralized-decentralized mismatch problem, and the credit assignment issue in deterministic policy learning. We proposed decomposed actor-critic methods (DOP) to address these problems. Theoretical analyses and empirical evaluations demonstrate that DOP can achieve stable and efficient multi-agent off-policy learning.

### ACKNOWLEDGMENTS

We would like to thank the anonymous reviewers for their insightful comments and helpful suggestions. This work is supported in part by Science and Technology Innovation 2030 – "New Generation Artificial Intelligence" Major Project (No. 2018AAA0100904), and a grant from the Institute of Guo Qiang, Tsinghua University.

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

# A MATHEMATICAL DETAILS FOR STOCHASTIC DOP

## A.1 DECOMPOSED CRITICS ENABLE TRACTABLE MULTI-AGENT TREE BACKUP

In Sec. 4.1.1, we propose to use tree backup (Precup et al., 2000; Munos et al., 2016) to carry out multi-agent off-policy policy evaluation. When a joint critic is used, calculating $\mathbb{E}_{\boldsymbol{\pi}}\left[Q_{tot}^{\phi}(\boldsymbol{\tau}, \cdot)\right]$ requires $O(|A|^n)$ steps of summation. To solve this problem, DOP uses a linearly decomposed critic, and it follows that:

$$
\begin{aligned}
\mathbb{E}_{\boldsymbol{\pi}}[Q_{tot}^{\phi}(\boldsymbol{\tau}, \boldsymbol{a})] &= \sum_{\boldsymbol{a}} \boldsymbol{\pi}(\boldsymbol{a}|\boldsymbol{\tau})Q_{tot}^{\phi}(\boldsymbol{\tau}, \boldsymbol{a}) = \sum_{\boldsymbol{a}} \boldsymbol{\pi}(\boldsymbol{a}|\boldsymbol{\tau})\left[\sum_i k_i(\boldsymbol{\tau})Q_i^{\phi_i}(\boldsymbol{\tau}, a_i) + b(\boldsymbol{\tau})\right] \\
&= \sum_{\boldsymbol{a}} \boldsymbol{\pi}(\boldsymbol{a}|\boldsymbol{\tau})\sum_i k_i(\boldsymbol{\tau})Q_i^{\phi_i}(\boldsymbol{\tau}, a_i) + \sum_{\boldsymbol{a}} \boldsymbol{\pi}(\boldsymbol{a}|\boldsymbol{\tau})b(\boldsymbol{\tau}) \\
&= \sum_i \sum_{a_i} \pi_i(a_i|\tau_i)k_i(\boldsymbol{\tau})Q_i^{\phi_i}(\boldsymbol{\tau}, a_i)\sum_{\boldsymbol{a}_{-i}} \boldsymbol{\pi}_{-i}(\boldsymbol{a}_{-i}|\boldsymbol{\tau}_{-i}) + b(\boldsymbol{\tau}) \\
&= \sum_i k_i(\boldsymbol{\tau})\mathbb{E}_{\pi_i}[Q_i^{\phi_i}(\boldsymbol{\tau}, \cdot)] + b(\boldsymbol{\tau}),
\end{aligned}
\tag{13}
$$

which means the complexity of calculating this expectation is reduced to $O(n|A|)$.

## A.2 STOCHASTIC DOP POLICY GRADIENTS

### A.2.1 ON-POLICY VERSION

In Sec. 4.1.1, we give the on-policy stochastic DOP policy gradients:

$$
g = \mathbb{E}_{\boldsymbol{\pi}}\left[\sum_i k_i(\boldsymbol{\tau})\nabla_{\theta_i}\log \pi_i(a_i|\tau_i; \theta_i)Q_i^{\phi_i}(\boldsymbol{\tau}, a_i)\right].
\tag{14}
$$

We now derive it in detail.

*Proof.* We use the aristocrat utility (Wolpert & Tumer, 2002) to perform credit assignment:

$$
\begin{aligned}
U_i(\boldsymbol{\tau}, a_i) &= Q_{tot}^{\phi}(\boldsymbol{\tau}, \boldsymbol{a}) - \sum_x \pi_i(x|\tau_i)Q_{tot}^{\phi}(\boldsymbol{\tau}, (x, \boldsymbol{a}_{-i})) \\
&= \sum_j k_j(\boldsymbol{\tau})Q_j^{\phi_j}(\boldsymbol{\tau}, a_j) - \sum_x \pi_i(x|\tau_i)\left[\sum_{j \neq i} k_j(\boldsymbol{\tau})Q_j^{\phi_j}(\boldsymbol{\tau}, a_j) + k_i(\boldsymbol{\tau})Q_i^{\phi_i}(\boldsymbol{\tau}, x)\right] \\
&= k_i(\boldsymbol{\tau})Q_i^{\phi_i}(\boldsymbol{\tau}, a_i) - k_i(\boldsymbol{\tau})\sum_x \pi_i(x|\tau_i)Q_i^{\phi_i}(\boldsymbol{\tau}, x) \\
&= k_i(\boldsymbol{\tau})\left[Q_i^{\phi_i}(\boldsymbol{\tau}, a_i) - \sum_x \pi_i(x|\tau_i)Q_i^{\phi_i}(\boldsymbol{\tau}, x)\right],
\end{aligned}
$$

It is worth noting that $U_i$ is independent of other agents' actions. Then, for the policy gradients, we have:

$$
\begin{aligned}
g &= \mathbb{E}_{\boldsymbol{\pi}}[\sum_i \nabla_\theta \log \pi_i(a_i|\tau_i)U_i(\boldsymbol{\tau}, a_i)] \\
&= \mathbb{E}_{\boldsymbol{\pi}}\left[\sum_i \nabla_\theta \log \pi_i(a_i|\tau_i)k_i(\boldsymbol{\tau})\left(Q_i^{\phi_i}(\boldsymbol{\tau}, a_i) - \sum_x \pi_i(x|\tau_i)Q_i^{\phi_i}(\boldsymbol{\tau}, x)\right)\right] \\
&= \mathbb{E}_{\boldsymbol{\pi}}\left[\sum_i \nabla_\theta \log \pi_i(a_i|\tau_i)k_i(\boldsymbol{\tau})Q_i^{\phi_i}(\boldsymbol{\tau}, a_i)\right].
\end{aligned}
$$

$\square$

### A.2.2 Off-policy version

In Appendix A.2, we derive the on-policy policy gradients for updating stochastic multi-agent policies. Similar to policy evaluation, using off-policy data can improve the sample efficiency with regard to policy improvement.

Using the linearly decomposed critic architecture, the off-policy policy gradients for learning stochastic policies are:

$$g = \mathbb{E}_{\boldsymbol{\beta}} \left[ \frac{\pi(\boldsymbol{\tau}, \boldsymbol{a})}{\beta(\boldsymbol{\tau}, \boldsymbol{a})} \sum_i k_i(\boldsymbol{\tau}) \nabla_\theta \log \pi_i(a_i|\tau_i; \theta_i) Q_i^{\phi_i}(\boldsymbol{\tau}, a_i) \right]. \tag{15}$$

*Proof.* The objective function is:

$$J(\theta) = \mathbb{E}_{\boldsymbol{\beta}} \left[ V_{tot}^{\boldsymbol{\pi}}(\boldsymbol{\tau}) \right].$$

Similar to Degris et al. (2012), we have:

$$
\begin{aligned}
\nabla_\theta J(\theta) &= \mathbb{E}_{\boldsymbol{\beta}} \left[ \frac{\pi(\boldsymbol{a}|\boldsymbol{\tau})}{\beta(\boldsymbol{a}|\boldsymbol{\tau})} \sum_i \nabla_\theta \log \pi_i(a_i|\tau_i) U_i(\boldsymbol{\tau}, a_i) \right] \\
&= \mathbb{E}_{\boldsymbol{\beta}} \left[ \frac{\pi(\boldsymbol{a}|\boldsymbol{\tau})}{\beta(\boldsymbol{a}|\boldsymbol{\tau})} \sum_i \nabla_\theta \log \pi_i(a_i|\tau_i) k_i(\boldsymbol{\tau}) A_i(\boldsymbol{\tau}, a_i) \right] \\
&= \mathbb{E}_{\boldsymbol{\beta}} \left[ \frac{\pi(\boldsymbol{a}|\boldsymbol{\tau})}{\beta(\boldsymbol{a}|\boldsymbol{\tau})} \sum_i \nabla_\theta \log \pi_i(a_i|\tau_i) k_i(\boldsymbol{\tau}) Q_i^{\phi_i}(\boldsymbol{\tau}, a_i) \right].
\end{aligned}
$$

$\square$

## B  Mathematical details for deterministic DOP

### B.1  Deterministic DOP policy gradient theorem

In Sec. 4.2.1, we give the following deterministic DOP policy gradients:

$$\nabla J(\theta) = \mathbb{E}_{\boldsymbol{\tau} \sim \mathcal{D}} \left[ \sum_i k_i(\boldsymbol{\tau}) \nabla_{\theta_i} \mu_i(\tau_i; \theta_i) \nabla_{a_i} Q_i^{\phi_i}(\boldsymbol{\tau}, a_i)|_{a_i = \mu_i(\tau_i; \theta_i)} \right]. \tag{16}$$

Now we present the derivation of this update rule.

*Proof.* Drawing inspirations from single-agent cases (Silver et al., 2014), we have:

$$
\begin{aligned}
\nabla J(\theta) &= \mathbb{E}_{\boldsymbol{\tau} \sim \mathcal{D}}[\nabla_\theta Q_{tot}^\phi(\boldsymbol{\tau}, \boldsymbol{a})] \\
&= \mathbb{E}_{\boldsymbol{\tau} \sim \mathcal{D}}[\sum_i \nabla_\theta k_i(\boldsymbol{\tau}) Q_i^{\phi_i}(\boldsymbol{\tau}, a_i)|_{a_i = \mu_i(\tau_i; \theta_i)}] \\
&= \mathbb{E}_{\boldsymbol{\tau} \sim \mathcal{D}}[\sum_i \nabla_\theta \mu_i(\tau_i; \theta_i) \nabla_{a_i} k_i(\boldsymbol{\tau}) Q_i^{\phi_i}(\boldsymbol{\tau}, a_i)|_{a_i = \mu_i(\tau_i; \theta_i)}].
\end{aligned}
$$

$\square$

## C  Theoretical justification for stochastic DOP policy improvement

In order to understand how DOP works despite the biased $Q^{\boldsymbol{\pi}}(\boldsymbol{\tau}, \boldsymbol{a})$ estimation, we provide some theoretical justification for the policy update. Unfortunately, a thorough analysis on deep neural network and TD-learning is too complex to be carried out. Thus, we make some assumptions for the mathematical proof. The following two subsections provide two different view points of theoretical understanding.

1. In the first view (Sec. C.1), we assume some mild assumptions on value evaluation which holds on a wide range of function class. In this way, we can prove a policy improvement theorem similar to Degris et al. (2012).

2. In the second view, we remove the MONOTONE condition from a practical point of view. We then prove that when the loss of value evaluation is minimized (individual critics output $Q_i^{\phi_i}(\boldsymbol{\tau}, a_i)$ to be a good estimate of $Q_i^{\boldsymbol{\pi}}(\boldsymbol{\tau}, a_i)$), the DOP gradients in Eq. 12 equal to Eq. 2 which is the standard gradient form.

## C.1 PROOF OF STOCHASTIC DOP POLICY IMPROVEMENT THEOREM

Inspired by previous work (Degris et al., 2012), we relax the requirement that $Q_{tot}^{\phi}$ is a good estimate of $Q_{tot}^{\boldsymbol{\pi}}$ and show that stochastic DOP still guarantees policy improvement.

First, we define

$$Q_i^{\boldsymbol{\pi}}(\boldsymbol{\tau}, a_i) = \sum_{\boldsymbol{a}_{-i}} \boldsymbol{\pi}_{-i}(\boldsymbol{a}_{-i}|\boldsymbol{\tau}_{-i}) Q_{tot}^{\boldsymbol{\pi}}(\boldsymbol{\tau}, \boldsymbol{a}), \quad A_i^{\boldsymbol{\pi}}(\boldsymbol{\tau}, a_i) = \sum_{\boldsymbol{a}_{-i}} \boldsymbol{\pi}(\boldsymbol{a}_{-i}|\boldsymbol{\tau}_{-i}) A_i^{\boldsymbol{\pi}}(\boldsymbol{\tau}, \boldsymbol{a}).$$

Directly analyzing the minimization of TD-error is challenging. To make it tractable, some works (Feinberg et al., 2018) simplify this analysis to an MSE problem. For the analysis of stochastic DOP, we adopt the same technique and formalize the critic's learning as the following problem:

$$L(\phi) = \sum_{\boldsymbol{a}, \boldsymbol{\tau}} p(\boldsymbol{\tau}) \pi(\boldsymbol{a}|\boldsymbol{\tau}) \left( Q_{tot}^{\boldsymbol{\pi}}(\boldsymbol{\tau}, \boldsymbol{a}) - Q_{tot}^{\phi}(\boldsymbol{\tau}, \boldsymbol{a}) \right)^2, \tag{17}$$

where $Q_{tot}^{\boldsymbol{\pi}}(\boldsymbol{\tau}, \boldsymbol{a})$ are the true values, which are fixed during optimization. In the following analysis, we assume distinct parameters for different $\boldsymbol{\tau}$. We first show that Fact 1 holds for a wide range of function class of $Q_i^{\phi_i}$. To this end, we first prove the following lemma.

**Lemma 1.** *Without loss of generality, we consider the following optimization problem:*

$$L_{\boldsymbol{\tau}}(\phi) = \sum_{\boldsymbol{a}} \boldsymbol{\pi}(\boldsymbol{a}|\boldsymbol{\tau}) \left( Q^{\boldsymbol{\pi}}(\boldsymbol{\tau}, \boldsymbol{a}) - f(\mathbf{Q}^{\phi}(\boldsymbol{\tau}, \boldsymbol{a})) \right)^2. \tag{18}$$

*Here, $f(\mathbf{Q}^{\phi}(\boldsymbol{\tau}, \boldsymbol{a})): \mathcal{R}^n \to \mathcal{R}$, and $\mathbf{Q}^{\phi}(\boldsymbol{\tau}, \boldsymbol{a})$ is a vector whose $i^{th}$ entry is $Q_i^{\phi_i}(\boldsymbol{\tau}, a_i)$. In DOP, $f$ satisfies that $\frac{\partial f}{\partial Q_i^{\phi_i}(\boldsymbol{\tau}, a_i)} > 0$ for any $i, a_i$.*

*If $\nabla_{\phi_i} Q_i^{\phi_i}(\boldsymbol{\tau}, a_i) \neq 0, \forall \phi_i, a_i$ it holds that:*

$$Q_i^{\boldsymbol{\pi}}(\boldsymbol{\tau}, a_i) \geq Q_i^{\boldsymbol{\pi}}(\boldsymbol{\tau}, a_i') \iff Q_i^{\phi_i}(\boldsymbol{\tau}, a_i) \geq Q_i^{\phi_i}(\boldsymbol{\tau}, a_i'), \quad \forall a_i, a_i'.$$

*Proof.* When the optimization converges, $\phi_i$ reaches a stationary point where $\nabla_{\phi_i} L_{\boldsymbol{\tau}}(\phi) = 0, \forall i$.

$$\pi_i(a_i|\tau_i) \sum_{\boldsymbol{a}_{-i}} \prod_{j \neq i} \pi_j(a_j|\tau_j) \left( Q_{tot}^{\boldsymbol{\pi}}(\boldsymbol{\tau}, \boldsymbol{a}) - f(\mathbf{Q}^{\phi}(\boldsymbol{\tau}, \boldsymbol{a})) \right) \left( -\frac{\partial f}{\partial Q_i^{\phi_i}(\boldsymbol{\tau}, a_i)} \right) \nabla_{\phi_i} Q_i^{\phi_i}(\boldsymbol{\tau}, a_i) = 0, \quad \forall a_i.$$

Since $\nabla_{\phi_i} Q_i^{\phi_i}(\boldsymbol{\tau}, a_i) \neq 0$, this implies that $\forall i, a_i$, we have

$$\sum_{\boldsymbol{a}_{-i}} \prod_{j \neq i} \pi_j(a_j|\tau_j) (Q_{tot}^{\boldsymbol{\pi}}(\boldsymbol{\tau}, \boldsymbol{a}) - f(\mathbf{Q}^{\phi}(\boldsymbol{\tau}, \boldsymbol{a}))) = 0$$

$$\Rightarrow \sum_{\boldsymbol{a}_{-i}} \boldsymbol{\pi}_{-i}(\boldsymbol{a}_{-i}|\boldsymbol{\tau}_{-i}) f(\mathbf{Q}^{\phi}(\boldsymbol{\tau}, \boldsymbol{a})) = Q_i^{\boldsymbol{\pi}}(\boldsymbol{\tau}, a_i)$$

We consider the function $q(\boldsymbol{\tau}, a_i) = \sum_{\boldsymbol{a}_{-i}} \boldsymbol{\pi}_{-i}(\boldsymbol{a}_{-i}|\boldsymbol{\tau}_{-i}) f(\mathbf{Q}^{\phi}(\boldsymbol{\tau}, \boldsymbol{a})))$, which is a function of $\mathbf{Q}^{\phi}$. Its partial derivative w.r.t $Q_i^{\phi_i}(\boldsymbol{\tau}, a_i)$ is:

$$\frac{\partial q(\boldsymbol{\tau}, a_i)}{\partial Q_i^{\phi_i}(\boldsymbol{\tau}, a_i)} = \sum_{\boldsymbol{a}_{-i}} \boldsymbol{\pi}_{-i}(\boldsymbol{a}_{-i}|\boldsymbol{\tau}_{-i}) \frac{\partial f(\mathbf{Q}^{\phi}(\boldsymbol{\tau}, \boldsymbol{a}))}{\partial Q_i^{\phi_i}(\boldsymbol{\tau}, a_i)} > 0$$

Therefore, if $Q_i^{\boldsymbol{\pi}}(\boldsymbol{\tau}, a_i) \geq Q_i^{\boldsymbol{\pi}}(\boldsymbol{\tau}, a_i')$, then any local minimal of $L_{\boldsymbol{\tau}}(\phi)$ satisfies $Q_i^{\phi_i}(\boldsymbol{\tau}, a_i) \geq Q_i^{\phi_i}(\boldsymbol{\tau}, a_i')$. $\hfill \square$

We argue that $\nabla_{\phi_i} Q_i^{\phi_i}(\boldsymbol{\tau}, a_i) \neq 0$ is a rather mild assumption and holds for a large range of function class of $Q_i^{\phi_i}$.

**Fact 3.** *[Formal Statement of Fact 1] For the following choices of $Q_i^{\phi_i}$:*

1. *Tabular expression of $Q_i^{\phi_i}$ which requires $Q(n|A||\boldsymbol{\tau}|)$ space;*

2. *Linear function class where $a_i$ are one-hot coded:*

$$Q_i^{\phi_i}(\boldsymbol{\tau}, a_i) = \phi_i \cdot \langle \boldsymbol{\tau}, a_i \rangle;$$

3. *2-layer neural networks ($\phi_i \neq \mathbf{0}$) with strictly monotonic increasing activation functions (e.g. tanh, leaky-relu).*

4. *Arbitrary $k$-layer neural networks whose activation function at the $(k-1)^{th}$ layer is sigmoid.*

*when value evaluation converges, $\forall \boldsymbol{\pi}$, $Q_i^{\phi_i}$ satisfies that*

$$Q_i^{\boldsymbol{\pi}}(\boldsymbol{\tau}, a_i) \geq Q_i^{\boldsymbol{\pi}}(\boldsymbol{\tau}, a_i') \iff Q_i^{\phi_i}(\boldsymbol{\tau}, a_i) \geq Q_i^{\phi_i}(\boldsymbol{\tau}, a_i'), \quad \forall \boldsymbol{\tau}, a_i, a_i'.$$

*Proof.* We need to prove that $\nabla_{\phi_i} Q_i^{\phi_i}(\boldsymbol{\tau}, a_i) \neq 0$. For brevity, we use $a_i^k$ to denote the $k^{\text{th}}$ element of the one-hot coding, and use $\phi_i^{t a_i^k}$ to denote the weight connecting the $t^{\text{th}}$ element of the upper layer and the $a_i^k$ element.

(1 & 2) For tabular expression and linear functions, $\forall a_i = k$ we have

$$\frac{\partial Q_i^{\phi_i}(\boldsymbol{\tau}, a_i)}{\partial \phi_i^{1 a_i^k}} = 1$$

(3) The 2-layer neural network can be written as $Q_i^{\phi_i}(\boldsymbol{\tau}, a_i) = W_2 \sigma(W_1(\boldsymbol{\tau}, a_i))$. Besides, we denote the hidden layer as $h$. Since $\phi_i \neq 0$, we consider some nonzero element $\phi_{1t,i}^{W_2}$. For the $k^{\text{th}}$ action, the gradient of the parameter $\phi_{tk,i}^{W_1}$ is

$$\frac{\partial Q_i^{\phi_i}(\boldsymbol{\tau}, a_i)}{\partial \phi_{tk,i}^{W_1}} = \phi_{1t,i}^{W_2} \sigma'(h_t) \neq 0, \quad \forall k$$

(4) Without loss of generality, we consider the last layer $\phi_{1t,i}^{W_k}$:

$$\frac{\partial Q_i^{\phi_i}(\boldsymbol{\tau}, a_i)}{\partial \phi_{1t,i}^{W_k}} = \sigma(h_t^{k-1}) > 0$$

These are the cases where $\nabla_{\phi_i} Q_i^{\phi_i} \neq 0$. Even when $\exists \phi_i, \nabla_{\phi_i} Q_i^{\phi_i} = 0$, such $\phi_i$ usually occupy only a small parameter space and happen with a small probability. As a result, we conclude that $\nabla_{\phi_i} Q_i^{\phi_i}(\boldsymbol{\tau}, a_i) \neq 0$ is a rather mild assumption. $\hfill \square$

Based on Fact 1, we are able to prove the policy improvement theorem for stochastic DOP. We will show that even without an accurate estimate of $Q_{tot}^{\boldsymbol{\pi}}$, the stochastic DOP policy updates can still improve the objective function $J(\boldsymbol{\pi}) = \mathbb{E}_{\boldsymbol{\pi}}[\sum_t \gamma^t r_t]$. We first prove the following lemma.

**Lemma 2.** *For two sequences $\{a_i\}, \{b_i\}, i \in [n]$ listed in an increasing order. If $\sum_i b_i = 0$, then $\sum_i a_i b_i \geq 0$.*

*Proof.* We denote $\bar{a} = \frac{1}{n}\sum_i a_i$, then $\sum_i a_i b_i = \bar{a}(\sum_i b_i) + \sum_i \tilde{a}_i b_i$ where $\sum_i \tilde{a}_i = 0$. Without loss of generality, we assume that $\bar{a}_i = 0, \forall i$. $j$ and $k$ which $a_j \leq 0, a_{j+1} \geq 0$ and $b_k \leq 0, b_{k+1} \geq 0$. Since $a, b$ are symmetric, we assume $j \leq k$. Then we have

$$
\begin{aligned}
\sum_{i\in[n]} a_i b_i &= \sum_{i\in[1,j]} a_i b_i + \sum_{i\in[j+1,k]} a_i b_i + \sum_{i\in[k+1,n]} a_i b_i \\
&\geq \sum_{i\in[j+1,k]} a_i b_i + \sum_{i\in[k+1,n]} a_i b_i \\
&\geq a_k \sum_{i\in[i+1,k]} b_i + a_{k+1} \sum_{i\in[k+1,n]} b_i
\end{aligned}
$$

As $\sum_{i\in[j+1,n]} b_i \geq 0$, we have $-\sum_{i\in[j+1,k]} b_i \leq \sum_{i\in[k+1,n]} b_i$.

Thus, $\sum_{i\in[n]} a_i b_i \geq (a_{k+1} - a_k)\sum_{i\in[k+1,n]} b_i \geq 0$. $\qquad\square$

Based on Fact 1 and Lemma 2, we prove the following proposition.

**Proposition 2.** *[Stochastic DOP policy improvement theorem] Under mild assumptions, for any pre-update policy $\boldsymbol{\pi}^o$ which is updated by Eq. 10 to $\boldsymbol{\pi}$, denote $\pi_i(a_i|\tau_i) = \pi_i^o(a_i|\tau_i) + \beta_{a_i,\boldsymbol{\tau}}\delta$, where $\delta > 0$ is a sufficiently small number. If it holds that $\forall\tau, a_i', a_i, Q_i^{\phi_i}(\boldsymbol{\tau}, a_i) \geq Q_i^{\phi_i}(\boldsymbol{\tau}, a_i') \iff \beta_{a_i,\boldsymbol{\tau}} \geq \beta_{a_i',\boldsymbol{\tau}}$ (MONOTONE condition, and $\phi_i$ is the parameters before update.), then we have*

$$J(\boldsymbol{\pi}) \geq J(\boldsymbol{\pi}^o), \tag{19}$$

*i.e., the joint policy is improved by the update.*

*Proof.* Under Fact 1, it follows that

$$Q_i^{\boldsymbol{\pi}^o}(\boldsymbol{\tau}, a_i) > Q_i^{\boldsymbol{\pi}^o}(\boldsymbol{\tau}, a_i') \iff \beta_{a_i,\boldsymbol{\tau}} \geq \beta_{a_i',\boldsymbol{\tau}}. \tag{20}$$

Since $J(\boldsymbol{\pi}) = \sum_{\boldsymbol{\tau}_0} p(\boldsymbol{\tau}_0)V_{tot}^{\boldsymbol{\pi}}(\boldsymbol{\tau}_0)$, it suffices to prove that $\forall\boldsymbol{\tau}_t, V_{tot}^{\boldsymbol{\pi}}(\boldsymbol{\tau}_t) \geq V_{tot}^{\boldsymbol{\pi}^o}(\boldsymbol{\tau}_t)$. We have:

$$
\begin{aligned}
\sum_{\boldsymbol{a}_t} \boldsymbol{\pi}(\boldsymbol{a}_t|\boldsymbol{\tau}_t)Q_{tot}^{\boldsymbol{\pi}^o}(\boldsymbol{\tau}_t, \boldsymbol{a}_t) &= \sum_{\boldsymbol{a}_t} \left(\prod_{i=1}^n \pi_i(a_i^t|\tau_i^t)\right) Q_{tot}^{\boldsymbol{\pi}^o}(\boldsymbol{\tau}_t, \boldsymbol{a}_t) \\
&= \sum_{\boldsymbol{a}_t} \left(\prod_{i=1}^n (\pi_i^o(a_i^t|\tau_i^t) + \beta_{a_i^t,\boldsymbol{\tau}_t}\delta)\right) Q_{tot}^{\boldsymbol{\pi}^o}(\boldsymbol{\tau}_t, \boldsymbol{a}_t) \\
&= V_{tot}^{\boldsymbol{\pi}^o}(\boldsymbol{\tau}_t) + \delta \sum_{i=1}^n \sum_{\boldsymbol{a}_t} \beta_{a_i^t,\boldsymbol{\tau}_t}\left(\prod_{j\neq i}\pi_j^o(a_j^t|\tau_j^t)\right) Q_{tot}^{\boldsymbol{\pi}^o}(\boldsymbol{\tau}_t, \boldsymbol{a}_t) + o(\delta) \\
&= V_{tot}^{\boldsymbol{\pi}^o}(\boldsymbol{\tau}_t) + \delta \sum_{i=1}^n \sum_{a_i^t} \beta_{a_i^t,\boldsymbol{\tau}_t} Q_i^{\boldsymbol{\pi}^o}(\boldsymbol{\tau}_t, a_i^t) + o(\delta). \tag{21}
\end{aligned}
$$

Since $\delta$ is sufficiently small, in the following analysis we omit $o(\delta)$. Observing that $\sum_{a_i} \pi_i(a_i|\tau_i) = 1, \forall i$, we get $\sum_{a_i} \beta_{a_i,\boldsymbol{\tau}} = 0$. Thus, by Lemma 2 and Eq. 21, we have

$$\sum_{\boldsymbol{a}_t} \boldsymbol{\pi}(\boldsymbol{a}_t|\boldsymbol{\tau}_t)Q_{tot}^{\boldsymbol{\pi}^o}(\boldsymbol{\tau}_t, \boldsymbol{a}_t) \geq V_{tot}^{\boldsymbol{\pi}^o}(\boldsymbol{\tau}_t). \tag{22}$$

Similar to the policy improvement theorem for tabular MDPs (Sutton & Barto, 2018) , we have

$$
\begin{aligned}
V_{tot}^{\boldsymbol{\pi}^o}(\boldsymbol{\tau}_t) &\leq \sum_{\boldsymbol{a}_t} \boldsymbol{\pi}(\boldsymbol{a}_t|\boldsymbol{\tau}_t)Q_{tot}^{\boldsymbol{\pi}^o}(\boldsymbol{\tau}_t, \boldsymbol{a}_t) \\
&= \sum_{\boldsymbol{a}_t} \boldsymbol{\pi}(\boldsymbol{a}_t|\boldsymbol{\tau}_t)\left(r(\boldsymbol{\tau}_t, \boldsymbol{a}_t) + \gamma \sum_{\boldsymbol{\tau}_{t+1}} p(\boldsymbol{\tau}_{t+1}|\boldsymbol{\tau}_t, \boldsymbol{a}_t)V_{tot}^{\boldsymbol{\pi}^o}(\boldsymbol{\tau}_{t+1})\right)
\end{aligned}
$$

$$\leq \sum_{\boldsymbol{a}_t} \boldsymbol{\pi}(\boldsymbol{a}_t|\boldsymbol{\tau}_t) \left( r(\boldsymbol{\tau}_t, \boldsymbol{a}_t) + \gamma \sum_{\boldsymbol{\tau}_{t+1}} p(\boldsymbol{\tau}_{t+1}|\boldsymbol{\tau}_t, \boldsymbol{a}_t) \left( \sum_{\boldsymbol{a}_{t+1}} \boldsymbol{\pi}(\boldsymbol{a}_{t+1}|\boldsymbol{\tau}_{t+1}) Q_{tot}^{\boldsymbol{\pi}^o}(\boldsymbol{\tau}_{t+1}, \boldsymbol{a}_{t+1}) \right) \right)$$

$$\leq \cdots$$

$$\leq V_{tot}^{\boldsymbol{\pi}}(\boldsymbol{\tau}_t).$$

This implies $J(\boldsymbol{\pi}) \geq J(\boldsymbol{\pi}^o)$ for each update.

Moreover, we verify that $\forall \tau, a_i', a_i, Q_i^{\phi_i}(\boldsymbol{\tau}, a_i) > Q_i^{\phi_i}(\boldsymbol{\tau}, a_i') \iff \beta_{a_i, \boldsymbol{\tau}} \geq \beta_{a_i', \boldsymbol{\tau}}$ (the MONOTONE condition) holds for any $\boldsymbol{\pi}$ with a tabular expression. For these $\boldsymbol{\pi}$, let $\pi_i(a_i|\tau_i) = \theta_{a_i, \boldsymbol{\tau}}$, then it holds that $\sum_{a_i} \theta_{a_i, \boldsymbol{\tau}} = 1$. Since the gradient of policy update can be written as:

$$\nabla_\theta J(\boldsymbol{\pi}_\theta) = \mathbb{E}_{d(\tau)} \left[ \sum_i k_i(\boldsymbol{\tau}) \nabla_\theta \log \pi_i(a_i|\tau_i; \theta_i) Q_i^{\phi_i}(\boldsymbol{\tau}, a_i) \right]$$

$$= \sum_\tau d(\tau) \sum_i k_i(\boldsymbol{\tau}) \nabla_{\theta_i} \pi(a_i|\tau_i) Q_i^{\phi_i}(\boldsymbol{\tau}, a_i)$$

$$= \sum_\tau d(\tau) \sum_i k_i(\boldsymbol{\tau}) \nabla_{\theta_i} \pi(a_i|\tau_i) A_i^{\phi_i}(\boldsymbol{\tau}, a_i)$$

where $d^\pi(\tau)$ is the occupancy measure w.r.t our algorithm. With a tabular expression, the update of each $\theta_{a_i, \boldsymbol{\tau}}$ is proportion to $\beta_{a_i, \boldsymbol{\tau}}$

$$\beta_{a_i, \boldsymbol{\tau}} \propto \frac{d\eta(\pi_\theta)}{d\theta_{a_i, \boldsymbol{\tau}}} = d(\boldsymbol{\tau}) A_i^{\phi_i}(\boldsymbol{\tau}, a_i)$$

Clearly, $\beta_{a_i', \boldsymbol{\tau}} \geq \beta_{a_i, \boldsymbol{\tau}} \iff Q_i^{\phi_i}(\boldsymbol{\tau}, a_i') \geq Q_i^{\phi_i}(\boldsymbol{\tau}, a_i)$. $\qquad\square$

## C.2 ANALYSIS WITHOUT MONOTONE CONDITION

For practical implementation of policy $\pi_i(a_i|\tau_i)$, the MONOTONE condition is too strong to be satisfied for all $\pi_i$. Analyzing the policy update when the condition is violated is difficult with only Fact 1 at hand. Therefore, it is beneficial to understand policy improvement without the MONOTONE condition.

To bypass the MONOTONE condition, we require a stronger property of the learnt $Q_i^{\phi_i}(\boldsymbol{\tau}, a_i)$ in addition to order preserving (Fact 1). Theorem 1 in Wang et al. (2020a) offers a closed form solution of additive decomposition and we restate it as the following lemma

**Lemma 3** (Restatement of Theorem 1 in Wang et al. (2020a)). *If we consider the solution of*

$$\arg\min_Q \sum_{(s,\mathbf{a}) \in \mathcal{S} \times \mathbf{A}} \boldsymbol{\pi}(\boldsymbol{a}|\boldsymbol{\tau}) \left( y(\boldsymbol{\tau}, \boldsymbol{a}) - \sum_{i=1}^n Q_i(\boldsymbol{\tau}, a_i) \right)^2,$$

$\forall i \in [n], \forall \boldsymbol{\tau}, \boldsymbol{a}$ *the individual action-value function* $Q_i(\boldsymbol{\tau}, a_i) =$

$$\mathbb{E}_{a_{-i} \sim \boldsymbol{\pi}_{-i}(\cdot|\boldsymbol{\tau}_{-i})} [y(\boldsymbol{\tau}, a_i, a_{-i})] - \frac{n-1}{n} \mathbb{E}_{\boldsymbol{a} \sim \pi(\cdot|\boldsymbol{\tau})} [y(\boldsymbol{\tau}, \boldsymbol{a})] + w_i(s), \tag{23}$$

*The residual term* $\mathbf{w}$ *is an arbitrary vector satisfying* $\forall s, \sum_{i=1}^n w_i(s) = 0$.

Based on this lemma, we can derive another proposition to theoretically justify the DOP architecture.

**Proposition 1.** *Suppose the function class expressed by* $Q_i^{\phi_i}(\boldsymbol{\tau}, a_i)$ *is sufficiently large (e.g. neural networks) and the following loss* $L(\phi)$ *is minimized*

$$L(\phi) = \sum_{\boldsymbol{a}, \boldsymbol{\tau}} p(\boldsymbol{\tau}) \boldsymbol{\pi}(\boldsymbol{a}|\boldsymbol{\tau}) (Q_{tot}^{\boldsymbol{\pi}}(\boldsymbol{\tau}, \boldsymbol{a}) - Q_{tot}^\phi(\boldsymbol{\tau}, \boldsymbol{a}))^2,$$

where $Q_{tot}^{\phi}(\boldsymbol{\tau}, \mathbf{a}) \equiv \sum_i k_i(\boldsymbol{\tau}) Q_i^{\phi_i}(\boldsymbol{\tau}, a_i) + b(\boldsymbol{\tau})$. *Then, we have*

$$g = \mathbb{E}_{\boldsymbol{\pi}} \left[ \sum_i \nabla_{\theta_i} \log \pi_i(a_i|\tau_i; \theta_i) Q^{\boldsymbol{\pi}}(\boldsymbol{\tau}, \boldsymbol{a}) \right]$$

$$= \mathbb{E}_{\boldsymbol{\pi}} \left[ \sum_i k_i(\boldsymbol{\tau}) \nabla_{\theta_i} \log \pi_i(a_i|\tau_i; \theta_i) Q_i^{\phi_i}(\boldsymbol{\tau}, a_i) \right],$$

*which means stochastic DOP policy gradients are the same as those calculated using centralized critics (Eq. 2). Therefore, policy improvement is guaranteed.*

*Proof.* For brevity, we denote $Q k_i^{\phi_i}(\boldsymbol{\tau}, a_i) = k(\boldsymbol{\tau}) Q_i^{\phi_i}(\boldsymbol{\tau}, a_i)$. Then $L(\phi)$ can be written as

$$L(\phi) = \sum_{\boldsymbol{a}, \boldsymbol{\tau}} p(\boldsymbol{\tau}) \pi(\boldsymbol{a}|\boldsymbol{\tau}) (Q_{tot}^{\boldsymbol{\pi}}(\boldsymbol{\tau}, \boldsymbol{a}) - \sum_i Q k_i^{\phi_i}(\boldsymbol{\tau}, a_i) - b(\boldsymbol{\tau}))^2$$

According to Lemma 3, when $L(\phi)$ is minimized, we have

$$Q k_i^{\phi_i}(\boldsymbol{\tau}, a_i) = Q_i^{\boldsymbol{\pi}}(\boldsymbol{\tau}, a_i) - \frac{n-1}{n} V^{\boldsymbol{\pi}}(\boldsymbol{\tau}) + w_i(s) - \frac{1}{n} b^*(\boldsymbol{\tau})$$

$$= Q_i^{\boldsymbol{\pi}}(\boldsymbol{\tau}, a_i) - w_i'(\boldsymbol{\tau})$$

Then

$$g = \mathbb{E}_{\boldsymbol{\pi}} \left[ \sum_i k_i(\boldsymbol{\tau}) \nabla_{\theta_i} \log \pi_i(a_i|\tau_i; \theta_i) Q_i^{\phi_i}(\boldsymbol{\tau}, a_i) \right]$$

$$= \mathbb{E}_{\boldsymbol{\pi}} \left[ \sum_i \nabla_{\theta_i} \log \pi_i(a_i|\tau_i; \theta_i) Q k_i^{\phi_i}(\boldsymbol{\tau}, a_i) \right]$$

$$= \mathbb{E}_{\boldsymbol{\pi}} \left[ \sum_i \nabla_{\theta_i} \log \pi_i(a_i|\tau_i; \theta_i) Q^{\boldsymbol{\pi}}(\boldsymbol{\tau}, \boldsymbol{a}) \right]$$

$\square$

Therefore, in expectation, stochastic DOP gradients are the same as those calculated using centralized critics (Eq. 2). We no longer require the MONOTONE condition to guarantee improvement of the policy update. Proposition 1 is another point of view to explain the performance guarantee of DOP despite its constrained critics.

## D    REPRESENTATIONAL CAPABILITY OF DETERMINISTIC DOP CRITICS

In Sec. 4.2.2, we present the following facts about deterministic DOP:

**Fact 2.** *Assume that $\forall \boldsymbol{\tau}, \boldsymbol{a}, \boldsymbol{a}' \in O_\delta(\boldsymbol{\tau})$, $\| \nabla_{\boldsymbol{a}} Q_{tot}^{\boldsymbol{\mu}}(\boldsymbol{\tau}, \boldsymbol{a}) - \nabla_{\boldsymbol{a}'} Q_{tot}^{\boldsymbol{\mu}}(\boldsymbol{\tau}, \boldsymbol{a}') \|_2 \le L \| \boldsymbol{a} - \boldsymbol{a}' \|_2$. The estimation error of a DOP critic can be bounded by $O(L\delta^2)$ for $\boldsymbol{a} \in O_\delta(\boldsymbol{\tau}), \forall \boldsymbol{\tau}$.*

We consider the Taylor expansion with Lagrange remainder of $Q_{tot}^{\boldsymbol{\mu}}(\boldsymbol{\tau}, \boldsymbol{a})$. Namely,

$$Q_{tot}^{\boldsymbol{\mu}}(\boldsymbol{\tau}, \boldsymbol{a}) = Q_{tot}^{\boldsymbol{\mu}}(\boldsymbol{\tau}, \boldsymbol{\mu}(\tau)) + \nabla_{\boldsymbol{a}} Q_{tot}^{\boldsymbol{\mu}}(\boldsymbol{\tau}, \boldsymbol{a})|_{\boldsymbol{a}=\boldsymbol{\mu}(\boldsymbol{\tau})} \cdot (\boldsymbol{a} - \boldsymbol{\mu}(\boldsymbol{\tau})) + \frac{1}{2} \nabla^2 Q_{tot}^{\boldsymbol{\mu}}(\boldsymbol{\tau}, \boldsymbol{a}_\varsigma) \| \boldsymbol{a} - \pi(\boldsymbol{\tau}) \|^2$$

Since $\forall \boldsymbol{a} \in O_\delta(\pi(\boldsymbol{\tau}))$, we have

$$|Q_{tot}^{\boldsymbol{\mu}}(\boldsymbol{\tau}, \boldsymbol{a}) - Q_{tot}^{\boldsymbol{\mu}}(\boldsymbol{\tau}, \boldsymbol{\mu}(\boldsymbol{\tau})) - \nabla_{\boldsymbol{a}} Q_{tot}^{\boldsymbol{\mu}}(\boldsymbol{\tau}, \boldsymbol{a})|_{\boldsymbol{a}=\boldsymbol{\mu}(\boldsymbol{\tau})} \cdot (\boldsymbol{a} - \boldsymbol{\mu}(\boldsymbol{\tau}))| \le \frac{1}{2} L \delta^2$$

Noticing that the first order Taylor expansion of $Q_{tot}^{\boldsymbol{\mu}}$ has the form $\sum_{[n]} k_i(\boldsymbol{\tau}) Q_i^{\phi}(\boldsymbol{\tau}, a_i) + b(\boldsymbol{\tau})$. Therefore, the optimal solution of the MSE problem in Eq. 17 under DOP critics has an error term less than $O(L\delta^2)$ for arbitrary sampling distribution $p(\boldsymbol{\tau}, \boldsymbol{a})$ of $\boldsymbol{a} \in O_\delta(\boldsymbol{\mu}(\boldsymbol{\tau}))$.

When Q values in the proximity of $\mu(\boldsymbol{\tau}), \forall \boldsymbol{\tau}$ is well estimated within a bounded error and $\delta \ll 1$, approximately, we have

$$|\frac{\partial Q_{tot}^{\boldsymbol{\mu}}(\boldsymbol{\tau}, \boldsymbol{a})}{\partial a_i} - \frac{\partial Q_{tot}^{\phi}(\boldsymbol{\tau}, \boldsymbol{a})}{\partial a_i}| \approx |\frac{Q_{tot}^{\boldsymbol{\mu}}(\boldsymbol{\tau}, a_{-i}, a_i + \delta) - Q_{tot}^{\boldsymbol{\mu}}(\boldsymbol{\tau}, \boldsymbol{a})}{\delta} - \frac{Q_{tot}^{\phi}(\boldsymbol{\tau}, a_{-i}, a_i + \delta) - Q_{tot}^{\phi}(\boldsymbol{\tau}, \boldsymbol{a})}{\delta}|$$

$$= |\frac{Q_{tot}^{\boldsymbol{\mu}}(\boldsymbol{\tau}, a_{-i}, a_i + \delta) - Q_{tot}^{\phi}(\boldsymbol{\tau}, a_{-i}, a_i + \delta)}{\delta} - \frac{Q_{tot}^{\boldsymbol{\mu}}(\boldsymbol{\tau}, \boldsymbol{a}) - Q_{tot}^{\phi}(\boldsymbol{\tau}, \boldsymbol{a})}{\delta}|$$

$$\sim O(L\delta)$$

# E    ALGORITHMS

In this section, we describe the details of our algorithms, as shown in Algorithm 1 and 2.

---
**Algorithm 1** Stochastic DOP
---

Initialize a critic network $Q^\phi$, actor networks $\pi_{\theta_i}$, and a mixer network $M^\psi$ with random parameters $\phi, \theta_i, \psi$.
Initialize target networks: $\phi' = \phi$, $\theta' = \theta$, $\psi' = \psi$
Initialize an off-policy replay buffer $\mathcal{D}_{\text{off}}$ and an on-policy replay buffer $\mathcal{D}_{\text{on}}$.
**for** $t = 1$ to $T$ **do**
    Generate a trajectory and store it in $\mathcal{D}_{\text{off}}$ and $\mathcal{D}_{\text{on}}$
    Sample a batch consisting of $N_1$ trajectories from $\mathcal{D}_{\text{on}}$
    Update decentralized policies using the gradients described in Eq. 10
    Calculate $\mathcal{L}^{\text{On}}(\phi)$
    Sample a batch consisting of $N_2$ trajectories from $\mathcal{D}_{\text{off}}$
    Calculate $\mathcal{L}^{\text{DOP-TB}}(\phi)$
    Update critics using $\mathcal{L}^{\text{On}}(\phi)$ and $\mathcal{L}^{\text{DOP-TB}}(\phi)$
    **if** $t \bmod d = 0$ **then**
        Update target networks: $\phi' = \phi$, $\theta' = \theta$, $\psi' = \psi$
    **end if**
**end for**

---

---
**Algorithm 2** Deterministic DOP
---

Initialize a critic network $Q^\phi$, actor networks $\mu_{\theta_i}$ and a mixer network $M^\psi$ with random parameters $\theta, \phi, \psi$
Initialize target networks: $\phi' = \phi$, $\theta' = \theta$, $\psi' = \psi$
Initialize replay buffer $\mathcal{D}$
**for** $t = 1$ to $T$ **do**
    Select action with exploration noise $\boldsymbol{a} \sim \boldsymbol{\mu}(\boldsymbol{\tau}) + \epsilon$, generate a transition and store the transition tuple in $\mathcal{D}$
    Sample $N$ transitions from $\mathcal{D}$
    Update the critic using the loss function described in Eq. 11
    Update decentralized policies using the gradients described in Eq. 12
    **if** $t \bmod d = 0$ **then**
        Update target networks: $\phi' = \alpha\phi + (1-\alpha)\phi'$, $\theta' = \alpha\theta + (1-\alpha)\theta'$, $\psi' = \alpha\psi + (1-\alpha)\psi'$
    **end if**
**end for**

---

# F    DOP WITH COMMUNICATION

Although DOP can solve many coordination problems, as shown by the comparison against IQL in Fig.6, its complete decomposition critic raises the concern that DOP can not deal with the miscoordination problem induced by highly uncertain and partial observable environments.

We use an example to illustrate the causes of miscoordination problems and argue that introducing communication into DOP can help address these problems. In **hallway** (Fig. 7(a)), two agents randomly start at states $a_1$ to $a_m$ and $b_1$ to $b_n$, respectively. Agents can observe their position and choose to move left, move right, or keep still at each timestep. Agents win and are rewarded 10 if they arrive at state $g$ simultaneously. Otherwise, if any agent arrives at $g$ earlier than the other, the team gets no reward, and the next episode begins. The horizon is set to $\max(m, n) + 10$ to avoid an infinite loop.

Without communication, one agent cannot know the position of its teammates, so it is difficult to coordinate actions. This explains why on *hallway* with $m=n=4$, the team can win only 25% of the games (Fig. 7(b)). Equipping DOP with communication can largely solve the problem – agents learn to communicate their positions and move left at $a_1$ or $b_1$ simultaneously. For communication, we

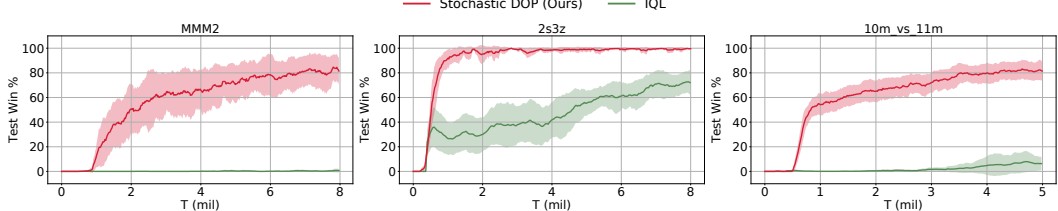

Figure 6: A decomposed critic can solve many coordination problems which can not be solved by IQL.

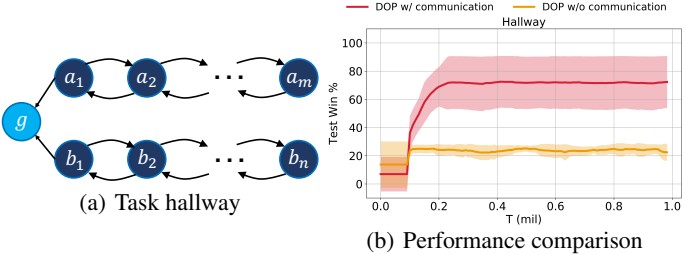

(a) Task hallway

(b) Performance comparison

Figure 7: A highly partial observable task. (a) Task **hallway**; (b) Performance of DOP with and without communication on *hallway* with $m=n=4$.

use the technique introduced by Wang et al. (2020e). Agents share a communication module, and messages are passed both between actors and individual Q-functions.

Such miscoordination problems are common in complex multi-agent tasks (Wang et al., 2020e). We believe introducing communication into DOP can help it solve a larger range of problems.

## G    BASELINE BY SAMPLING

One problem of existing MAPG methods is the CDM issue, which describes the large variance in policy gradients caused by the influence of other agents' actions introduced through the joint critic. Another technique that is frequently used to reduce the variance in policy gradients in the single-agent RL literature is by using baselines (Sutton & Barto, 2018). In this section, we investigate whether using baselines can effectively reduce variance in multi-agent settings.

We start from centralized critics. COMA uses a baseline where local actions are marginalized. Since the variance and performance of COMA have been discussed in Sec. 5, we omit it here and study the baseline where actions of all agents are marginalized. In multi-agent settings, the calculation of this baseline requires computing an expectation over the joint action space, which is generally intractable. To solve this problem, we estimate the expectation by sampling.

We compare stochastic DOP, COMA, and *On-Policy DOP* against this method, which we call **Regular Critics with Baseline**. Results are shown in Fig. 8. We can see that *Regular Critics with Baseline* performs better than COMA. However, *Regular Critics with Baseline* performs worse than *On-Policy DOP*. These results indicate that a linearly decomposed critic can reduce variance in policy gradients more efficiently.

## H    INFRASTRUCTURE, ARCHITECTURE, AND HYPERPARAMETERS

Experiments are carried out on NVIDIA P100 GPUs and with fixed hyper-parameter settings, which are described in the following sections.

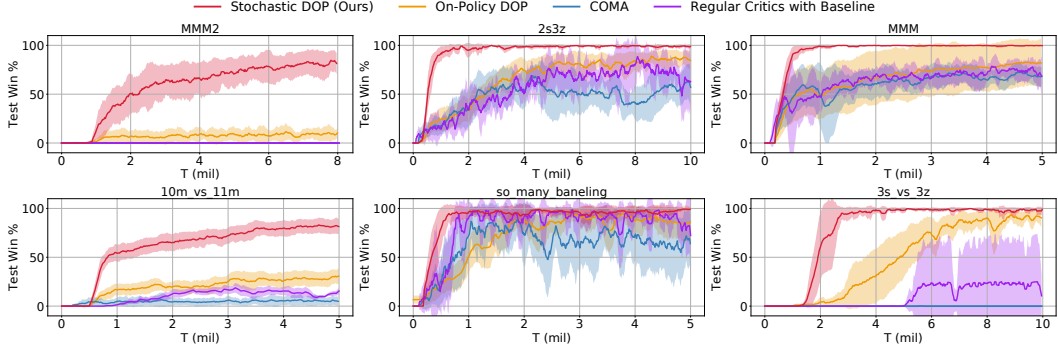

Figure 8: Using baselines where actions of all other agents are marginalized within a centralized critic is more efficient than COMA, but less efficient than a decomposed critic.

## H.1  STOCHASTIC DOP

In stochastic DOP, each agent has a neural network to approximate its local utility. The local utility network consists of two 256-dimensional fully-connected layers with ReLU activation. Since the critic is not used when execution, we condition local Q networks on the global state $s$. The output of the local utility networks is $Q_i^{\phi_i}(\boldsymbol{\tau}, \cdot)$ for each possible local action, which are then linearly combined to get an estimate of the global Q value. The weights and bias of the linear combination, $k_i$ and $b$, are generated by linear networks conditioned on the global state $s$. $k_i$ is enforced to be non-negative by applying absolute activation at the last layer. We then divide $k_i$ by $\sum_i k_i$ to scale $k_i$ to $[0, 1]$.

The local policy network consists of three layers, a fully-connected layer, followed by a 64 bit GRU, and followed by another fully-connected layer that outputs a probability distribution over local actions. We use ReLU activation after the first fully-connected layer.

For all experiments, we set $\kappa = 0.5$ and use an off-policy replay buffer storing the latest 5000 episodes and an on-policy buffer with a size of 32. We run 4 parallel environments to collect data. The optimization of both the critic and actors is conducted using RMSprop with a learning rate of $5 \times 10^{-4}$, $\alpha$ of 0.99, and with no momentum or weight decay. For exploration, we use $\epsilon$-greedy with $\epsilon$ annealed linearly from 1.0 to 0.05 over $500k$ time steps and kept constant for the rest of the training. Mixed batches consisting of 32 episodes sampled from the off-policy replay buffer and 16 episodes sampled from the on-policy buffer are used to train the critic. For training actors, we sample 16 episodes from the on-policy buffer each time. The framework is trained on fully unrolled episodes. The learning rates for the critic and actors are set to $1 \times 10^{-4}$ and $5 \times 10^{-4}$, respectively. And we use 5-step decomposed multi-agent tree backup. All experiments on StarCraft II use the default reward and observation settings of the SMAC benchmark.

## H.2  DETERMINISTIC DOP

The critic network structure of deterministic DOP is similar to that of stochastic DOP, except that local actions are part of the input in deterministic DOP. For actors, we use a fully-connected forward network with two 64-dimensional hidden layers with ReLU activation, and the output of actors is a local action. We use an off-policy replay buffer storing the latest $10000$ transitions, from which $1250$ transitions are sampled each time to train the critic and actors. The learning rates of both the critic and actors are set to $5 \times 10^{-3}$. To reduce variance in the updates of actors, we update the actors and target networks only after 2 updates to the critic, as proposed by Fujimoto et al. (2018). We also use this technique of delaying policy update in all the baselines. For all the algorithms, we run a single environment to collect data, because we empirically find it more sample efficient than parallel environments in the MPE benchmark. RMSprop with a learning rate of $5 \times 10^{-4}$, $\alpha$ of 0.99, and with no momentum or weight decay is used to optimize the critic and actors, which is the same as in stochastic DOP.

# I  RELATED WORKS

Cooperative multi-agent reinforcement learning provides a scalable approach to learning collaborative strategies for many challenging tasks (Vinyals et al., 2019; Berner et al., 2019; Samvelyan et al., 2019; Jaderberg et al., 2019) and a computational framework to study many problems, including the emergence of tool usage (Baker et al., 2020), communication (Foerster et al., 2016; Sukhbaatar et al., 2016; Lazaridou et al., 2017; Das et al., 2019), social influence (Jaques et al., 2019), and inequity aversion (Hughes et al., 2018). Recent work on role-based learning (Wang et al., 2020c; 2021) introduces the concept of division of labor into multi-agent learning and grounds MARL into more realistic applications.

Centralized learning of joint actions can handle coordination problems and avoid non-stationarity. However, the major concern of centralized training is scalability, as the joint action space grows exponentially with the number of agents. The coordination graph (Guestrin et al., 2002b;a) is a promising approach to achieve scalable centralized learning, which exploits coordination independencies between agents and decomposes a global reward function into a sum of local terms. Zhang & Lesser (2011) employ the distributed constraint optimization technique to coordinate distributed learning of joint action-value functions. Sparse cooperative Q-learning (Kok & Vlassis, 2006) learns to coordinate the actions of a group of cooperative agents only in the states where such coordination is necessary. These methods require the dependencies between agents to be pre-supplied. To avoid this assumption, value function decomposition methods directly learn centralized but factorized global Q-functions. They implicitly represent the coordination dependencies among agents by the decomposable structure (Sunehag et al., 2018; Rashid et al., 2018; Son et al., 2019; Wang et al., 2020e). The stability of multi-agent off-policy learning is a long-standing problem. Foerster et al. (2017); Wang et al. (2020a) study this problem in value-based methods. In this paper, we focus on how to achieve efficient off-policy policy-based learning. Our work is complementary to previous work based on multi-agent policy gradients, such as those regarding multi-agent multi-task learning (Teh et al., 2017; Omidshafiei et al., 2017) and multi-agent exploration (Wang et al., 2020d).

Multi-agent policy gradient algorithms enjoy stable convergence properties compared to value-based methods (Gupta et al., 2017; Wang et al., 2020a) and can extend MARL to continuous control problems. COMA (Foerster et al., 2018) and MADDPG (Lowe et al., 2017) propose the paradigm of centralized critic with decentralized actors to deal with the non-stationarity issue while maintaining decentralized execution. PR2 (Wen et al., 2019) and MAAC (Iqbal & Sha, 2019) extend the CCDA paradigm by introducing the mechanism of recursive reasoning and attention, respectively. Another line of research focuses on fully decentralized actor-critic learning (Macua et al., 2017; Zhang et al., 2018; Yang et al., 2018; Cassano et al., 2018; Suttle et al., 2019; Zhang & Zavlanos, 2019). Different from the setting of this paper, agents have local reward functions and full observation of the true state in these works.

