# OpenReview forum: "DOP: Off-Policy Multi-Agent Decomposed Policy Gradients"
_ICLR.cc/2021/Conference — ICLR 2021 Poster_

### Official Review · AnonReviewer1 · 2020-10-28
**Value decomposition and off-policy critic training for multi-agent RL with good experimental performance**

**Rating:** 7
**Confidence:** 4

**Review:**

In the context centralized training distributed execution in cooperative multi-agent reinforcement learning (MARL), the paper proposes an architecture to learn a decomposed action value function expressed as a weighted sum of the agent's individual functions (plus an additional weight). Those weights are themselves learned and depend on the observed history. Thanks to this decomposition, gradients can be decomposed over each agent. The authors propose to use a combination of off-policy (using tree backup) and on-policy (using TD(\lambda) methods for estimating the decomposed critic. They formulate both a deterministic and stochastic decomposed policy gradients, which are analyzed theoretically to some extent and evaluated experimentally.

PROS

The paper contributes to the active effort of designing more efficient MARL algorithms. The authors introduce the idea of value decomposition, which was investigated first in the value-based methods, to the actor-critic scheme. The experimental results suggest that the proposed combination of value decomposition and off-policy critic training has a good performance.

The writing of the paper is clear. The identification of the issue of centralized-decentralized mismatch and how it is tackled by the proposed methods are useful and interesting.


CONS

The formulation of Prop. 1 is hard to understand. What is Q_i^{\phi_i}? Is it the value function wrt \pi or \pi^0? Besides, how do we ensure that the strict inequality about Q_i^{\phi_i} holds?

Some theoretical analyses (e.g., A3 or C) are only loosely related to the actual proposed method, although I agree that a direct analysis would be difficult to conduct.

---

> ### Author Response · Authors · 2020-11-19
> **Thanks for the insightful review. We improve the theoretical parts (Prop. 1 and the corresponding appendixes).**
>
> We thank the reviewer for the thoughtful review.
>
> **Q1**: About Proposition 1. "What is $Q_i^{\phi_i}$? Is it the value function wrt $\pi$ or $\pi^o$? Besides, how do we ensure that the strict inequality about $Q_i^{\phi_i}$ holds?"
>
> **A1**: (1) $Q_i^{\phi_i}$ is the value function for $\pi^o$. We have updated the statement of this proposition to make this point clear. (2) In Appendix C.1, we prove that the inequality holds when $\mathbf{\pi}$ has a tabular expression. We agree with the reviewer that this condition for the inequality to be held is strong, so we relax this condition by giving Proposition 2 (page 6, proved on page 19-20) in the updated version. This proposition proves that for parameterized $\pi_i$ which does not satisfy the original condition, policy improvement can still be guaranteed as long as the function class expressed by local Q-functions ($Q_i^{\phi_i}$) is sufficiently large (e.g. neural networks) and the loss of critic training is minimized.
>
> **Q2**: "Some theoretical analyses (e.g., A3 or C) are only loosely related to the actual proposed method."
>
> **A2**: We have removed Appendix A.3 and improved Appendix C. Appendix C now contains formal statements and proof for the following two conclusions.
> - (1) Fact 1 holds for the Q functions of the following function classes: (i) Functions with a tabular expression; (ii) The linear function class where actions are one-hot coded; (iii) 2-layer neural networks with strictly monotonic increasing activation functions (e.g., tanh, leaky-relu). (iv) Arbitrary $k$-layer neural networks whose activation function at the $(k-1)^{\text{th}}$ layer is sigmoid.
> - (2) Policy improvement guarantee still holds under mild assumptions about critics, without the original strict condition of Proposition 1. This result also addresses your first concern.

---

### Official Review · AnonReviewer3 · 2020-10-28
**More clarity needed as problem discussed is not well motivated**

**Rating:** 3
**Confidence:** 5

**Review:**

This paper focuses on the problem of multi-agent reinforcement learning (MARL) for CTDE scenario which is well studied in recent literature.  The work discusses shortcomings of actor-critic methods for MARL and proposes a solution using linearly factored critic. The paper is somewhat difficult to read and can be made better by deferring the details about previous methods to appendix. However my main concern is with the problem of centralized-decentralized mismatch (CDM) motivated in the paper and its proposed solution itself.

1. How exactly is a regular critic bad? As such a critic is supposed to be "true" to the policy, the requirement of decentralization has little bearing on the variance of policy gradients. Gradient noise increases with number of agents irrespective of whether there is centralized or decentralized execution.

2. The so called problem of CDM seems rather redundant (see 1 above), for example the authors say in page 3, 3rd para from bottom in line 3 that if the critic expectation under policy is negative, then individual policy performance is hurt. Such problem can easily be fixed using baselines, see Sutton and Barto, 2018 for example.

3. How is a linear factored critic compatible with an arbitrary joint policy? In general this not true and requires many strong assumptions, see for ex. Bhatnagar, 2009. While the authors acknowledge this, bypassing the actual complexity for modelling a joint critic with a linear one will in general render it insufficient to model inter-agent interactions. This puts into serious question, whether coordination is required in the experiment domains in the first place and if the performance improvement is just coming due to a biased but albeit easier to learn critic.

4. There are some unsupported claims which need better explanation like "This becomes problematic because a negative feedback loop is created, in which the joint critic is affected by the suboptimality of agent i, which disturbs policy updates of other agents" How is that so? The updates in principle can affect the policies of already suboptimal agents, which might fix them?

5. "Learning the decomposed critic implicitly realizes multi-agent credit assignment, because the individual critic provides credit
information for each agent to improve its policy in the direction of increasing the global expected return" again how so? claims like this need to be well supported.

6. Expectations are usually sampled so in principle even the $O(|A|^n)$ can be estimated with fewer samples incurring some variance, it might not be necessary to bias the critic drastically for this.

7. The authors need to shed more light on when the precondition $Q_i(\tau, a_i) > Q_i(\tau, a_i') \iff \beta_{a_i, \tau}\geq \beta_{a_i', \tau}$ in Prop. 1 holds beyond tabular settings. It seems a rather strong assumption to hold for all trajectory and
$O(|A|^n)$ inputs. Right now it seems rather grab bag to show policy improvement.

8. Why isn't comparison on SC2 done against more recent baselines like QTRAN, MAVEN, ROMA etc.?

---

> ### Author Response · Authors · 2020-11-19
> **Part 1: How exactly is a regular critic bad and the CDM issue.**
>
> We thank the reviewer for the review and comments.
>
> **Q1**: "How exactly is a regular critic bad?"
>
> **A1**: A regular critic has three drawbacks: (1) It does not well support off-policy learning for discrete action spaces. To calculate policy gradients, we need to estimate $Q_{tot}^{\mathbf\pi}$ in stead of $Q_{tot}^{*}$. Estimating $Q_{tot}^{\mathbf\pi}$ from off-policy data using a regular critic faces major challenges: (i) if we use importance sampling, the variance grows with the number of agents; (ii) if we use tree backup, we need to calculate an expectation ($E_{\mathbf{a} \sim \mathbf{\pi}}[Q_{tot}^{\mathbf{\pi}}(\mathbf{\tau}, \mathbf{a})]$) over the joint action space. In our experiments (Sec. 5.2.1, ablation study `DOP with common tree backup`), we show that estimating this expectation by sampling hurts performance. In contrast, the expectation can be calculated in linear time with accuracy when using a DOP critic (please refer to the comparison between DOP and *DOP with common tree backup* in Fig. 3 on page 7). (2) The policy gradients calculated using a regular critic suffer from large variance due to the influence of other agents' actions. (We will discuss this point in detail after we list all three shortcomings and in Q2.) (3) In continuous action spaces, it is largely unclear how to achieve credit assignment when using a regular critic. (We discuss how DOP achieves credit assignment in Q5.)
>
> Now we explain why a regular critic results in large variance in policy gradients by addressing the concerns of the reviewer.
> - "As such a critic is supposed to be "true" to the policy, the requirement of decentralization has little bearing on the variance of policy gradients."
> *Answer*: Although policy gradient induced by a regular critic is true in expectation, it suffers from large variance. Different from the single-agent case, we discuss the variance caused by the other agents' actions in our paper. Specifically, in existing multi-agent policy gradients (MAPG) methods, decentralized policies are conditioned on local actions, but policy gradients depend on regular critics, and thus are conditioned on all agents' actions, which can cause large variance (Eq. 4). In contrast, by decomposing the critic, DOP gradients are free from the influence of other agents' actions and thus can reduce variance.
>
> In Fig. 2, we empirically show the variance of gradients of different methods. We can see that a regular critic (COMA and MADDPG) induces much larger variance than DOP. (The reviewer thinks the CDM issue can be solved by baselines. However, after using baselines, the advantage still depends on other agents' actions. We provide more experiments and discussions about baselines in Q2.)
>
> - "Gradient noise increases with number of agents irrespective of whether there is centralized or decentralized execution. "
>
> *Answer*: In our paper, we discuss the decomposition of critics, which is not related to or influenced by whether there is decentralized or centralized execution. Both DOP and previous MAPG methods learn decentralized policies and execute in a decentralized manner. As for gradient noise, we empirically show that DOP can reduce gradient variance (Fig. 2-left).
>
> **Q2**: "CDM seems rather redundant. Such problem can easily be fixed using baselines."
>
> **A2**: CDM is caused by other agents' actions, and using baselines cannot get rid of the influence of these actions (because the advantage still depends on other agents' actions). For example, COMA already uses a baseline, but still suffers from much larger variance than DOP (Fig. 2). Furthermore, we show performance of COMA in Fig. 3. On the maps like `so_many_banelings`, COMA degenerates after learning a nearly optimal policy. This observation is in line with CDM -- the sub-optimality of individual policies can exacerbate each other through the variance induced by a regular critic.
>
> In the reviewer's defense, a COMA baseline only marginalizes the action of one agent. Therefore, as suggested by the citation ([Sutton and Barto, 2018]), we study another baseline where actions of all agents are marginalized. We add a new appendix (Appendix G) to analyze the effect of such a baseline. In multi-agent settings, calculating this baseline requires computing an expectation over the joint action space, which is generally intractable (3570$G$ summations on the map `MMM2`). We thus estimate the expectation by sampling, and call this method `Regular Critics with Baseline`. In Fig. 8 on page 23, we compare *Regular Critics with Baseline* with DOP, and we can see that DOP performs significantly better. To eliminate the influence of off-policy learning, we further omit the off-policy learning component of DOP and find that *Regular Critics with Baseline* underperforms *On-Policy DOP* on all tasks. (Another interesting observation is that *Regular Critics with Baseline* outperforms COMA.) These observations indicate that decomposing the critic is an efficient way to avoid CDM.

---

> > ### Author Response · Authors · 2020-11-19
> > **Part 2: Linearly decomposed critics and inter-agent interactions.**
> >
> > **Q3**: "How is a linear factored critic compatible with an arbitrary joint policy?" & "DOP is insufficient to model inter-agent interactions. Whether coordination is required in the experiment domains in the first place and if the performance improvement is just coming due to a biased but albeit easier to learn critic."
> >
> > **A3**: Although the critic is linearly decomposed, it is trained by using the global reward, which induces coordinated behaviors among agents. Moreover, in Proposition 2 of the updated version of the paper, we show that a linearly factored critic can guarantee policy improvement with respect to the expected global return. (The condition is relatively mild: we require the function class expressed by local Q-functions ($Q_i^{\phi_i}$) is sufficiently large, e.g., neural networks, and the critic loss is minimized. Detailed proof can be found in Appendix C.2.)
> >
> > Therefore, for a large range of tasks, a linearly factored critic is sufficient to model inter-agent interactions. For example, the SC2 tasks we test in the original paper requires coordination, which is confirmed by the fact that IQL (independent Q-learning) cannot learn well in these tasks as shown in Fig. 6 on page 22, and DOP can significantly outperform IQL.
> >
> > However, like other fully decentralized execution methods (VDN, QMIX, and COMA), DOP may not be sufficient to model interactions among agents in highly uncertain or partial observable environments. In Appendix F of the updated paper, we show such an example. We also find that this problem can be solved by introducing a communication module into DOP. We carry out experiments in Appendix F (Fig. 7 on page 22) to support this claim.

---

> > > ### Author Response · Authors · 2020-11-19
> > > **Part 3: Question 4-8**
> > >
> > > **Q4**: Explain "This becomes problematic because a negative feedback loop is created, in which the joint critic is affected by the suboptimality of agent $i$, which disturbs policy updates of other agents."
> > >
> > > **A4**: Before this sentence, we explain how the suboptimality of other agents negatively effect the updates of agent $i$ though the regular critic. By this sentence, we mean that after agent $i$ is effected negatively by the variance of the gradients, the suboptimality in its policy can similarly propagate to other agents again though the joint critic. In this way, the sub-optimality of individual policies will exacerbate each other, which is problematic. An example is shown in Fig. 3 (`so_many_banelings`), where COMA policy degrades after learning a nearly optimal policy.
> > >
> > > **Q5**: Explain "Learning the decomposed critic implicitly realizes multi-agent credit assignment, because the individual critic provides credit information for each agent to improve its policy in the direction of increasing the global expected return."
> > >
> > > **A5**: This claim restates Eq. 10 in natural language. Eq. 10 gives the gradients $\nabla_\theta J(\mathbf{\pi})$, which is "the direction of increasing the global expected return", and the right side depends on local Q-functions ($Q_{i}^{\phi_i}$) instead of $Q_{tot}$. This means that agent can improve the global expected return by only considering it local Q functions. Learning such local Q functions is the aim of credit assignment (as in VDN, QMIX, etc.).
> > >
> > > **Q6**: "Expectations are usually sampled so in principle even the $O(|A|^n)$ can be estimated with fewer samples incurring some variance, it might not be necessary to bias the critic drastically for this."
> > >
> > > **A6**: In the paper, we design two ablations to show sampling is not efficient in estimating the expectation over the joint action space.
> > >
> > > - (1) `DOP with common tree backup` (Fig. 4 on page 8). The difference between this ablation and DOP is that the expectation required by tree backup (Eq. 3) is estimated by 200 samples. We see that this ablation performs worse than *On-Policy DOP* which does not use off-policy data. This observation indicates that sampling does not estimate the off-policy learning target accurately. In contrast, DOP can quickly and stably converge using a similar number of operations (200 summations).
> > >
> > > - (2) `Regular critics with baseline`, as discussed in detail in Q2.
> > >
> > > Based on these results, we conclude that sampling is not efficient in calculating the expectation with few samples. We hypothesize that this is because the landscape of Q functions can be sophisticated in complex multi-agent tasks.
> > >
> > >
> > > **Q7**: "The authors need to shed more light on when the precondition $Q_i(\tau, a_i) > Q_i(\tau, a_i') \iff \beta_{a_i, \tau}\geq \beta_{a_i', \tau}$ in Prop. 1 holds beyond tabular settings."
> > >
> > > **A7**: We provide a new proposition (Proposition 2 on page 6 in the updated paper) to address your concern. We prove that policy improvement guarantee is still held without this precondition, as along as the function class represented by local Q-functions ($Q_i^{\phi_i}$) is sufficiently large (e.g. neural networks) and the critic loss is minimized. We provide a detailed statement of this proposition on page 6, and its proof can be found in Appendix C.2 (page 19-20) of the revised paper.
> > >
> > > **Q8**: "Why isn't comparison on SC2 done against more recent baselines like QTRAN, MAVEN, ROMA etc.?"
> > >
> > > **A8**: In the updated version, we compare DOP against QTRAN, MAVEN, and ROMA. Results are shown in  Fig. 3. In addition to these baselines, we also show the comparison against two other recent papers: NDQ and QPLEX in the same figure. DOP outperforms all these algorithms with significantly reduced variance across different seeds and tasks.

---

### Official Review · AnonReviewer2 · 2020-10-28
**Superlative work**

**Rating:** 9
**Confidence:** 4

**Review:**

This works motivates the use of a factorized critic for multi-agent policy gradient.   The technique is well-motivated, and the exposition anticipates and answers readers' likely concerns.   The experiment section is well-organized, supports the paper's major claims, and is empirically compelling.

The policy improvement claims in section 4.1.2 are initially unintuitive, but ultimately are intelligible as an agent-block-coordinate local optimality statement.  However this reviewer is not clear on the quality of these local optima (i.e., when do we get "trapped"?).  For example, is it possible to design a task where the local optima are all very poor?   Of course, the experiment section indicates many benchmark tasks are amenable to this decomposition; but perhaps reasoning about this would help in (re)defining multi-agent problems to encourage success, e.g., it would be interesting if adding actions that communicate information directly between agents mitigates the local optima problem.

---

> ### Author Response · Authors · 2020-11-19
> **Thanks for the inspiring review. We update policy improvement claims in section 4.2.1 and provide experimental results regarding communication.**
>
> We thank the reviewer for the thoughtful review and inspiring comments.
>
> We update Proposition 1 in Sec. 4.1.2 by relaxing its condition (Proposition 2, on page 6, proved in Appendix C.2). We prove that policy improvement can still be guaranteed as long as the function class expressed by local Q-functions ($Q_i^{\phi_i}$) is sufficiently large (e.g. neural networks) and the loss of critic training is minimized. We then answer your questions regarding communication.
>
> **Q1**: "It would be interesting if adding actions that communicate information directly between agents mitigates the local optima problem."
>
> **A1**: There are two kinds of communication in multi-agent settings. Action coordination by intention propagation and the communication of local observations. (1) For the first kind of communication, we agree with the reviewer that it can help DOP jump out of local optimal. For example, we can introduce deep coordination graph into DOP, which will help coordinate the action selections. (2) Secondly, in Appendix F, we show that communication of local observations can help DOP solve a larger range of problems. There (page 21-22), we use an example to show that the fully decomposed structure can lead to miscoordination problems in highly uncertain and partial observable environments, and that introducing a communication module into DOP can largely help address the problem.

---

### Official Review · AnonReviewer4 · 2020-10-31
**Review of DOP**

**Rating:** 7
**Confidence:** 3

**Review:**

Summary:

The paper proposed a simple but powerful idea of assuming a linear decomposition structure of the centralized critic into individual critics. It demonstrated 2 existing problems in the baseline MAPG agents (COMA, MADDPG), namely "centralized-decentralized mismatch" problem and credit assignment issues, and offers a theoretically motivated solution with carefully conducted experiment results including ablation studies as well as comparison with sota gradient-based and value-based algorithms.

##########################################################################

Reasons for score:

The paper is very well written. Its proposal is simple but theoretically motivated to solve existing problems in MAPG algorithms. The experiment results support its claims and provides additional insightful analysis. Therefore I recommend "accept". However, its theoretical justification is weakened by assuming a tabular case of pi and Q functions which is rarely the case in practice. Furthermore, in Proposition 1, the conditioning just above Eq. 11 looks quite particular and I'm not sure how general it is satisfied in practice. Last but not least, it would be nice to see comparisons with more recent value decomposition algorithms such as NDQ and QTRAN/QPLEX. Overall I gave it a score 7.

##########################################################################

Pros:

1. The paper is organized in very clear fashion. It is well motivated by existing problems and provides a simple and effective solution.
2. It conducted clever experimental analysis that made the advantages of DOP very obvious and understandable.
3. Its method of linear decomposition of the central critic into individual critics is simple yet powerful. The paper gave adequate theoretical support to this method.

##########################################################################

Cons:

1. Both the tabular assumption on functions pi and Q, and especially the conditioning of Proposition 1 weaken the theoretical guarantee that the algorithm will converge to local optima.
2. The paper did not compare with more recent value decomposition algorithms such as NDQ and QTRAN/QPLEX. It's unclear that the baselines are the SOTA results on the chosen domains.

##########################################################################

Questions during rebuttal period:

 Please address the two points in "Cons" above.

---

> ### Author Response · Authors · 2020-11-19
> **Thanks for the insightful comments. We relax the condition of Proposition 1 and extend it to cases beyond tabular functions. We also provide comparisons against more related works.**
>
> We thank the reviewer for the insightful and inspiring comments.
>
> **Q1**: "Both the tabular assumption on functions pi and Q, and especially the conditioning of Proposition 1 weaken the theoretical guarantee that the algorithm will converge to local optima."
>
> **A1**: In the updated version, we analyze cases beyond the tabular function class. Proof of the following conclusions is included in Appendix C of the updated version.
>
> (1) We prove that Fact 1 holds for the Q functions of the following function classes: (i) Functions with a tabular expression; (ii) The linear function class where actions are one-hot coded; (iii) 2-layer neural networks with strictly monotonic increasing activation functions (e.g. tanh, leaky-relu). (iv) Arbitrary $k$-layer neural networks whose activation function at the $(k-1)^{\text{th}}$ layer is sigmoid. Detailed statements and proof can be found in Appendix C.1.
>
> (2) In Proposition 2 on page 6 of the updated paper, we relax the condition of Proposition 1 and extend it to a larger range of function classes. We prove that, without the original condition of Proposition 1, stochastic DOP can still guarantee policy improvement as along as the function class represented by local Q-functions ($Q_i^{\phi_i}$) is sufficiently large (e.g. neural networks) and the critic loss can be minimized. Detailed statement, proof, and assumptions of this proposition can be found in Appendix C.2.
>
> **Q2**: "The paper did not compare with more recent value decomposition algorithms such as NDQ and QTRAN/QPLEX."
>
> **A2**: We compare DOP against NDQ, QTRAN, and QPLEX in the updated version. Results are shown in Fig. 3 on page 7. Additionally, we provide comparisons against the state-of-the-art multi-agent role-based method (ROMA) and multi-agent exploration method (MAVEN) in the same figure. DOP outperforms these baselines by a large margin.

---

### Decision · Program_Chairs · 2021-01-07
**Final Decision**

**Decision:**

Accept (Poster)

**Comment:**

The paper presents a decomposition of the value function in the context of CCDA.

Most reviewers find this paper clear and well written, although one reviewer suggests to change the paper structure.

The method presented in this paper is simple and well justified by a theoretical section. Experiments on several domains, including Starcraft 2 micro-management tasks, are supporting the claims of that section. After some reviewers pointed out that the tabular setup is not useful in practice, the authors have extended the empirical and theoretical results to a more general setup.

Some reviewers point out that some theoretical results may not be directly related to the experimental findings. In particular, reviewer 3 does not support a central claim of the paper, and find that CDM is misleading and not provably representing the core problem.
In general, reviewer 3 does not support acceptance of this paper, but I still believe this paper should be accepted based on the other reviews (clearly in favour of acceptance). I hope that the authors and reviewer 3 will be able to further discuss and reach understanding, which hopefully should lead to fruitful results.